# TAYLORNET: A TAYLOR-DRIVEN GENERIC NEURAL ARCHITECTURE

## ABSTRACT

In this work, we propose the Taylor Neural Network (TaylorNet), a generic neural architecture that parameterizes Taylor polynomials using DNNs without non-linear activation functions. The key challenges of developing TaylorNet lie in: (i) mitigating the curse of dimensionality caused by higher-order terms, and (ii) improving the stability of model training. To overcome these challenges, we first adopt Tucker decomposition to decompose the higher-order derivatives in Taylor expansion parameterized by DNNs into low-rank tensors. Then we propose a novel reducible TaylorNet to further reduce the computational complexity by removing more redundant parameters in the hidden layers. In order to improve training accuracy and stability, we develop a new Taylor initialization method. Finally, the proposed models are evaluated on a broad spectrum of applications, including image classification, natural language processing (NLP), and dynamical systems. The results demonstrate that our proposed Taylor-Mixer, which replaces MLP and activation layers in the MLP-Mixer with Taylor layer, can achieve comparable accuracy on image classification, and similarly on sentiment analysis in NLP, while significantly reducing the number of model parameters. More importantly, our method can explicitly learn and interpret some dynamical systems with Taylor polynomials. Meanwhile, the results demonstrate that our Taylor initialization can significantly improve classification accuracy compared to Xavier and Kaiming initialization.

## 1 INTRODUCTION

This paper proposes a generic neural architecture, called TaylorNet, that parameterizes Taylor polynomials using deep neural networks. It can be employed to a variety of application domains, including image classification, dynamical systems, and natural language processing (NLP). Importantly, the proposed method *does not use non-linear activation functions*, thus promoting interpretability of DNNs in some applications, such as dynamical systems.

This work is motivated by a growing popularity of physics-guided machine learning (ML) (Jia et al., 2021; Daw et al., 2017), which integrates physical priors into neural networks. Thus, it endows neural networks with the ability to generalize to new domains better. As a result, physics guided ML has been widely applied to a variety of areas, such as dynamical systems (Cranmer et al., 2020; Lusch et al., 2018; Greydanus et al., 2019), quantum mechanics (Schütt et al., 2017), and climate changes (Kashinath et al., 2021; Pathak et al., 2022). However, existing methods based on DNNs are either tailored to solve certain specific problems, such as PDEs (Li et al., 2020; Raissi et al., 2017) and dynamics prediction (Greydanus et al., 2019; Lusch et al., 2018; Wang et al., 2019), or hard to explain the results. Hence, the question is, can we develop a generic interpretable neural architecture that can be used in a wide range of machine learning domains?

In this paper, we develop a novel Taylor-driven neural architecture, called TaylorNet, that parameterizes Taylor polynomials using DNNs *without non-linear activation functions*, as shown in Fig. 1. The proposed TaylorNet is able to generalize to a wide spectrum of ML tasks, ranging from computer vision and dynamical systems to NLP. However, developing TaylorNet poses two main challenges. First, the computational complexity of Taylor polynomial parameterized by DNNs grows exponentially as the polynomial order increases. Second, its higher-order terms often lead to training instability. To deal with these challenges, we first adopt Tucker decomposition to decompose the higher-order

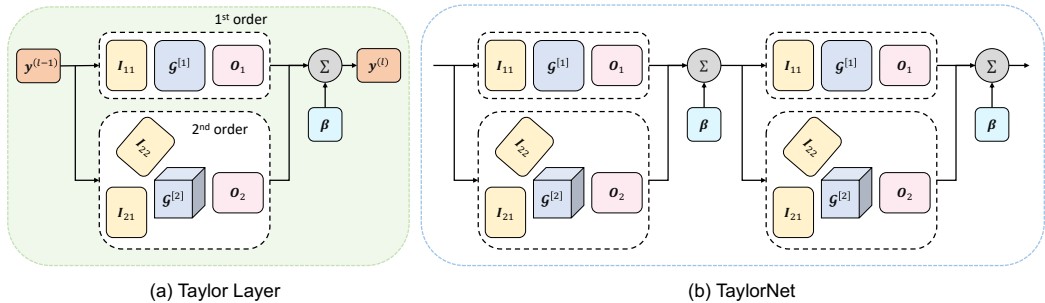

Figure 1: (a) Architecture of Taylor Layer of order 2 using Tucker decomposition; (b) TaylorNet consists of $N$ Taylor layers of order 2.

tensors in TaylorNet into a set of low-rank tensors (Malik & Becker, 2018; Kolda & Bader, 2009). To further reduce its computational complexity, we propose a reducible TaylorNet that removes more redundant learnable parameters in the hidden layers. In order to show the generalization of our architecture, we propose a Taylor-Mixer that uses Taylor layers in place of both the MLP layers and activation functions in the MLP-Mixer (Tolstikhin et al., 2021). Then a new Taylor initialization method is developed to improve the stability and accuracy of the model.

Finally, we evaluate the proposed Taylor-Mixer and TaylorNet in a variety of applications, including image classification, dynamical system, and NLP. The results show that our Taylor-Mixer can achieve comparable accuracy to the MLP-Mixer on image classification while exhibiting a considerable reduction of model parameters. The proposed TaylorNet also explicitly learns and interprets the dynamics of two classic physical systems with polynomials. Besides, our method can also be applied to NLP. The evaluation results on sentiment analysis demonstrate a competitive accuracy to the recently proposed pNLP-Mixer (Fusco et al., 2022). Meanwhile, our Taylor initialization can reach an accuracy that is over 10% higher than the Xavier and Kaiming initialization methods for the proposed TaylorNet (Glorot & Bengio, 2010; He et al., 2015).

In summary, our contributions include: 1) we design the TaylorNet, a novel neural architecture without activation functions that can generalize to a broad spectrum of ML domains, 2) we then propose a reducible TaylorNet that remarkably reduces the number of learning parameters, 3) a new Taylor initialization method is proposed to stabilize model training, 4) we develop a Taylor-Mixer that replaces both the MLP layers and activation functions in MLP-Mixer with Taylor layers, which can achieve comparable accuracy to the MLP-Mixer on image classification and sentiment analysis, and 5) Our approach can explicitly learn and explain some dynamical systems with polynomials, making way for interpretable ML.

## 2 RELATED WORK

**Polynomial Neural Networks.** We summarize the significant difference between the proposed TaylorNet and existing Polynomial Neural Networks. Earlier work Nikolaev & Iba (2006) mainly adopted Group Method of Data Handling to learn partial descriptors in polynomial neural networks. Then a follow-up pi-sigma network Shin & Ghosh (1991) was developed to filter the input features using predefined basis. These methods, however, fail to scale to high-dimensional data Chrysos et al. (2020). Recently, researchers designed $\Pi$-Nets Chrysos et al. (2020; 2022) that parameterizes the polynomial functions using deep neural networks and tensor decomposition. However, the performance of $\Pi$-Nets will degrade in larger networks. In addition, $\Pi$-Net can be viewed as a special case of TaylorNet at expansion point 0 since its adopted CP-decomposition is a special case of Tucker decomposition. Furthermore, we develop a novel Taylor initialization to improve the training stability while $\Pi$-Net does not. Our initialization method is different from the initialization paradigm for Tensorial Convolutional Neural Networks Pan et al. (2022).

**Related Work on Taylor Series.** Some recent studies developed Taylor-based neural networks. For example, TaylorSwiftNet Pourheydari et al. (2021) was developed for time-series prediction, but it is not applicable to high-dimensional classification. Recently, Mao et al Mao et al. (2022) developed Phy-Taylor to learn the physical dynamics based on partial physics knowledge, but it suffers from

the curse of dimensionality. Different from these methods, it can be used in a wide spectrum of application domains without using activation functions. Moreover, it can interpret some dynamical systems using Taylor polynomials.

**Learning Dynamics.** Some researchers developed physics-based DNNs to learn the dynamics of physical systems. For example, Lusch et al. designed the Koopman operator (Geneva & Zabaras, 2022; Lusch et al., 2018) that maps the non-linear dynamics into a linear Koopman representation space for predicting the future states of dynamical systems. Recent studies proposed Hamiltonian and Lagrangian neural networks (Cranmer et al., 2020) that strictly follow conservation laws. However, these methods are designed for specific problems, and they are hard to apply to other domains, such as computer vision and natural language processing (NLP).

**Tensor Decomposition.** Tensor decomposition (Kolda & Bader, 2009) aims to represent high-dimensional tensors using multilinear operations over the factor matrices or tensors. Some popular tensor decomposition methods include CP decomposition (Carroll & Chang, 1970), Tucker decomposition (Tucker, 1966), tensor train (TT) decomposition (Oseledets, 2011), and tensor ring (TR) decomposition (Zhao et al., 2016). Thanks to their ability to reduce computational complexity without breaking out data structure, tensor decomposition techniques are increasingly being widely used in machine learning applications (Wu et al., 2020; Pan et al., 2022; Qiu et al., 2021). Inspired by prior work, we adopt tensor decomposition to deal with the curse of dimensionality in our TaylorNet.

## 3 PRELIMINARIES

**Notations**. We summarize the main notations used throughout this paper in Appendix A. Regarding mathematical symbols, scalars are denoted by normal letters, e.g. $x$ and $y$, while vectors are denoted by lowercase boldface letters, e.g. $\boldsymbol{x}$. In addition, matrices are denoted by uppercase boldface letters, e.g. $\boldsymbol{X}$, while tensors are denoted by calligraphic letters, e.g. $\boldsymbol{\mathcal{X}}$ and $\boldsymbol{\mathcal{W}}$.

**Taylor Polynomial.** Taylor polynomial is able to approximate non-linear smooth functions given an arbitrary compact Hausdorff space according to Stone–Weierstrass theorem (Stone, 1948). According to Taylor's theorem (Thomas et al., 2005), for a vector-valued multivariate function $\boldsymbol{f} : \mathbb{R}^d \to \mathbb{R}^o$, its Taylor polynomial at a point $\boldsymbol{x} = \boldsymbol{x}_0$ can be expressed as

$$\boldsymbol{f}(\boldsymbol{x}) \approx \sum_{k=0}^{N} \frac{1}{k!} \left[ \sum_{j=0}^{d} \left( \Delta x_j \frac{\partial}{\partial x_j} \right) \right]^k \boldsymbol{f} \Big|_{\boldsymbol{x}_0}, \tag{1}$$

where $\boldsymbol{x} \in \mathbb{R}^d$, $\boldsymbol{x}_0 \in \mathbb{R}^d$, $\Delta x_j = x_j - x_{j,0}$, $j = 1, \ldots, d$. It can also be written as the following tensor form (Chrysos et al., 2020),

$$\boldsymbol{f}(\boldsymbol{x}) \approx \boldsymbol{f}(\boldsymbol{x}_0) + \sum_{k=1}^{N} \left( \boldsymbol{\mathcal{W}}^{[k]} \prod_{j=2}^{k+1} \bar{\times}_j \Delta \boldsymbol{x} \right), \tag{2}$$

where $\bar{\times}_m$ denotes *mode-m vector product*, $\Delta \boldsymbol{x} = \boldsymbol{x} - \boldsymbol{x}_0$ and $\boldsymbol{\mathcal{W}}^{[k]} \in \mathbb{R}^{o \prod_{m=1}^{k} \times d}$ are *scaled higher-order derivatives* of $\boldsymbol{f}$ at $\boldsymbol{x} = \boldsymbol{x}_0$. However, the problem of Taylor polynomial is that its computational complexity grows exponentially as the order increases, making it hard to be applied to high-dimensional data.

**Tucker Decomposition.** Tucker decomposition aims to decompose a tensor into a small core tensor multiplied by a set of matrices along the corresponding mode (Tucker, 1966; Kolda & Bader, 2009). In essence, Tucker decomposition can be viewed as a higher-order principal component analysis. Let $\boldsymbol{\mathcal{X}}$ be $N$-way tensors, then its Tucker decomposition is given by

$$\boldsymbol{\mathcal{X}} = \boldsymbol{\mathcal{G}} \times_1 \boldsymbol{A}^{(1)} \times_2 \cdots \times_N \boldsymbol{A}^{(N)}, \tag{3}$$

where $\boldsymbol{\mathcal{G}}$ is a core tensor and $\boldsymbol{A}^{(n)}$ $(n = 1, \ldots, N)$ are the factor matrices. According to mode-$n$ unfolding (Kolda & Bader, 2009), Eq. 3 can be expressed as the following matricized form:

$$\boldsymbol{X}_{(n)} = \boldsymbol{A}^{(n)} \boldsymbol{G}_{(n)} \left( \boldsymbol{A}^{(N)} \otimes \cdots \otimes \boldsymbol{A}^{(n+1)} \otimes \boldsymbol{A}^{(n-1)} \otimes \cdots \otimes \boldsymbol{A}^{(1)} \right)^{\top}, \tag{4}$$

where $\boldsymbol{X}_{(n)}$ and $\boldsymbol{G}_{(n)}$ are matrices that mean the mode-$n$ matricization of the tensor $\boldsymbol{\mathcal{X}}$ and $\boldsymbol{\mathcal{G}}$ and $\otimes$ denotes Kronecker product.

## 4  PROPOSED METHOD

In this section, we first introduce a lightweight Taylor Neural Network using Tucker decomposition. As an extension, we propose a reducible TaylorNet to further improve the computational efficiency by removing redundant trainable parameters in the middle layers. In order to stabilize the model training process and improve accuracy, a novel Taylor initialization method is developed in this work. Moreover, we present the connection between TaylorNet and some other existing neural networks.

### 4.1  TAYLOR NEURAL NETWORKS

Since mapping function $\boldsymbol{f}$ in Eq. 2 is unknown and needs to be learned by deep neural networks, we cannot calculate the derivatives $\boldsymbol{\mathcal{W}}^{[k]}$ of $\boldsymbol{f}$ directly. To deal with this issue, this work adopts deep neural networks to parameterize the Taylor polynomial in Eq. 2, where $\boldsymbol{f}(\boldsymbol{x}_0)$ and $\boldsymbol{\mathcal{W}}^{[k]}$ ($k = 1, \ldots, N$) are *learnable parameters* during model training. However, the computational complexity of tensor $\boldsymbol{\mathcal{W}}^{[k]}$ grows exponentially, $\mathcal{O}(d^k)$, as the polynomial order $k$ increases. To overcome this issue, Tucker decomposition is adopted in this work. According to Eq. 3, the scaled derivatives $\boldsymbol{\mathcal{W}}^{[k]}$ can be decomposed into

$$\boldsymbol{\mathcal{W}}^{[k]} = \boldsymbol{\mathcal{G}}^{[k]} \times_1 \boldsymbol{O}_k \times_2 \boldsymbol{I}_{k1} \cdots \times_{k+1} \boldsymbol{I}_{kk} = \boldsymbol{\mathcal{G}}^{[k]} \times_1 \boldsymbol{O}_k \prod_{j=1}^{k} \times_{j+1} \boldsymbol{I}_{kj} \tag{5}$$

where $\boldsymbol{\mathcal{G}}^{[k]} \in \mathbb{R}^{r_{\text{out},k} \prod_{j=1}^{k} \times r_{\text{in},k,j}}$ is the core tensor, $\boldsymbol{I}_{kj} \in \mathbb{R}^{d \times r_{\text{in},k,j}}$ ($j = 1, \ldots, k$) and $\boldsymbol{O}_k \in \mathbb{R}^{o \times r_{\text{out},k}}$ are input and output factor matrices, respectively. Here we use $r_{\text{in},k,j}$ and $r_{\text{out},k}$ to represent the Tucker ranks corresponding to the $j$-th input and output dimension in the $k$-th-order term of the Taylor polynomial.

Substituting Eq. 5 into 2, the $k$-th term of Taylor polynomial can be written as

$$\boldsymbol{\mathcal{W}}^{[k]} \prod_{j=2}^{k+1} \bar{\times}_j \Delta \boldsymbol{x} = \boldsymbol{\mathcal{W}}^{[k]} \prod_{j=2}^{k+1} \times_j \Delta \boldsymbol{x}^\top = \boldsymbol{\mathcal{G}}^{[k]} \times_1 \boldsymbol{O}_k \left( \prod_{i=1}^{k} \times_{i+1} \boldsymbol{I}_{ki} \right) \left( \prod_{j=1}^{k} \times_{j+1} \Delta \boldsymbol{x}^\top \right) \tag{6}$$

Based on commutative law and associative property in mode-$n$ product (Kolda & Bader, 2009), Eq. 6 can be reformulated as

$$\boldsymbol{\mathcal{W}}^{[k]} \prod_{j=2}^{k+1} \bar{\times}_j \Delta \boldsymbol{x} = \boldsymbol{\mathcal{G}}^{[k]} \times_1 \boldsymbol{O}_k \left[ \prod_{j=1}^{k} \times_{j+1} \left( \Delta \boldsymbol{x}^\top \boldsymbol{I}_{kj} \right) \right] \in \mathbb{R}^o. \tag{7}$$

Please refer to the detailed transformation in Appendix. B.

However, to our knowledge, the current deep learning frameworks (e.g, Pytorch and TensorFlow) do not support batch-size-based mode-$n$ product in Eq. 7. Fortunately, since the result of Eq. 7 is a vector, according to mode-$n$ unfolding as illustrated in Eqs. 3 and 4, we can convert it into a matricized form as follows.

$$\boldsymbol{\mathcal{W}}^{[k]} \prod_{j=2}^{k+1} \bar{\times}_j \Delta \boldsymbol{x} = \boldsymbol{O}_k \boldsymbol{G}_k \left[ \left( \boldsymbol{I}_{kk}^\top \Delta \boldsymbol{x} \right) \otimes \cdots \otimes \left( \boldsymbol{I}_{k1}^\top \Delta \boldsymbol{x} \right) \right], \tag{8}$$

Finally, substituting the above Eq. 8 into Taylor polynomial in Eq. 1, resulting in a lightweight $N$-th-order Taylor layer as follows.

$$\boldsymbol{f}(\boldsymbol{x}) = \boldsymbol{\beta} + \sum_{k=1}^{N} \boldsymbol{O}_k \boldsymbol{G}_k \left[ \left( \boldsymbol{I}_{kk}^\top \Delta \boldsymbol{x} \right) \otimes \cdots \otimes \left( \boldsymbol{I}_{k1}^\top \Delta \boldsymbol{x} \right) \right], \tag{9}$$

where $\boldsymbol{\beta} = \boldsymbol{f}(\boldsymbol{x}_0)$, $\boldsymbol{O}_k$, $\boldsymbol{G}_k$, and $\boldsymbol{I}_{kj}^\top$ ($k = 1, \ldots, N$; $j = 1, \ldots, k$) are learnable parameters.

After that, we can stack $L$ Taylor layers with a $N$-th order expansion to construct a new neural network, referred as the TaylorNet. According to Eq. 9 above, the output of the $l$-th layer in our TaylorNet with a $N$-th-order expansion can be written as

$$\boldsymbol{y}^{(l)} = \boldsymbol{\beta}^{(l)} + \sum_{k=1}^{N} \boldsymbol{O}_k^{(l)} \boldsymbol{G}_k^{(l)} \left\{ \left[ \left( \boldsymbol{I}_{kk}^{(l)} \right)^\top \boldsymbol{y}^{(l-1)} \right] \otimes \cdots \otimes \left[ \left( \boldsymbol{I}_{k1}^{(l)} \right)^\top \boldsymbol{y}^{(l-1)} \right] \right\}, \tag{10}$$

where $\boldsymbol{y}^{(l)} \in \mathbb{R}^{d^{(l+1)}}$ is the output of the $l$-th layer and $\boldsymbol{y}^{(0)} = \Delta \boldsymbol{x} \in \mathbb{R}^{d^{(1)}}$, $d^{(1)} = d$. Here $d^{(l)}$ is the input dimension of the $l$-th layer. In addition, $\boldsymbol{\beta}^{(l)} \in \mathbb{R}^{d^{(l+1)}}$, $\boldsymbol{O}_k^{(l)} \in \mathbb{R}^{d^{(l+1)} \times r_{\text{out},k}^{(l)}}$, $\boldsymbol{G}_k^{(l)} \in \mathbb{R}^{r_{\text{out},k}^{(l)} \times \prod_{j=1}^k r_{\text{in},k,j}^{(l)}}$, $\boldsymbol{I}_{kj}^{(l)} \in \mathbb{R}^{d^{(l)} \times r_{\text{in},k,j}^{(l)}}$ $(k = 1, \ldots, N; j = 1, \ldots, k)$ are learnable parameters of the $l$-th Tucker Taylor layer.

Fig. 1 shows the overall framework of the proposed TaylorNet. In this paper, we use the Taylor layer with *a second order expansion*, since it is able to mitigate the overfitting problem and also reduce the number of trainable parameters in DNNs.

**Remark 4.1.** *The computational complexity of the $k$-th-order term in Taylor layer decreases from $\mathcal{O}(od^k)$ to $\mathcal{O}(r_{\text{out},k} \prod_{j=1}^k r_{\text{in},k,j} + or_{\text{out},k} + d \sum_{j=1}^k r_{\text{in},k,j})$, where $d$ and $o$ denote the dimension of the input and the output. When $o$ and $d$ are much larger than the rank of a core tensor in Tucker decomposition, the number of parameters will be reduced by orders of magnitude.*

## 4.2 REDUCIBLE TAYLORNET AND TAYLOR-MIXER

**Reducible TaylorNet**. We further propose a reducible TaylorNet, called R-TaylorNet, to reduce the number of trainable parameters of TaylorNet. The basic idea is to use a single matrix as the new trainable parameter to replace the original product of the current layer's output factor matrix and the next layer's input factor matrix, namely, compositing two consecutively multiplying parameter matrices $\boldsymbol{O}_k^{(l)}$ and $\boldsymbol{I}_{kj}^{(l+1)}$ into a single parameter matrix. Below, we will theoretically derive the R-TaylorNet.

According to the block multiplication of matrices, Eq. 10 can be rewritten as

$$\boldsymbol{y}^{(l)} = \boldsymbol{\beta}^{(l)} + \left[ \boldsymbol{O}_1^{(l)} \, \boldsymbol{O}_2^{(l)} \, \cdots \, \boldsymbol{O}_N^{(l)} \right] \begin{bmatrix} \boldsymbol{G}_1^{(l)} \boldsymbol{z}_{11}^{(l)} \\ \boldsymbol{G}_2^{(l)} \left[ \boldsymbol{z}_{22}^{(l)} \otimes \boldsymbol{z}_{21}^{(l)} \right] \\ \vdots \\ \boldsymbol{G}_N^{(l)} \left[ \boldsymbol{z}_{NN}^{(l)} \otimes \cdots \otimes \boldsymbol{z}_{N1}^{(l)} \right] \end{bmatrix}, \tag{11}$$

where $\boldsymbol{z}_{kj}^{(l)} = \left( \boldsymbol{I}_{kj}^{(l)} \right)^\top \boldsymbol{y}^{(l-1)} \in \mathbb{R}^{r_{\text{in},k,j}^{(l)}}$, we call it hidden features of the $l$-th layer in this work.

In order to further simplify the above equation, we define the following notations:

$$\boldsymbol{O}^{(l)} \stackrel{\text{def}}{=} \left[ \boldsymbol{O}_1^{(l)} \, \boldsymbol{O}_2^{(l)} \, \cdots \, \boldsymbol{O}_N^{(l)} \right] \in \mathbb{R}^{d^{(l+1)} \times \sum_{k=1}^N r_{\text{out},k}^{(l)}},$$

$$\boldsymbol{h}\left( \boldsymbol{z}_{11}^{(l)}, \ldots, \boldsymbol{z}_{NN}^{(l)} \right) \stackrel{\text{def}}{=} \begin{bmatrix} \boldsymbol{G}_1^{(l)} \boldsymbol{z}_{11}^{(l)} \\ \boldsymbol{G}_2^{(l)} \left[ \boldsymbol{z}_{22}^{(l)} \otimes \boldsymbol{z}_{21}^{(l)} \right] \\ \vdots \\ \boldsymbol{G}_N^{(l)} \left[ \boldsymbol{z}_{NN}^{(l)} \otimes \cdots \otimes \boldsymbol{z}_{N1}^{(l)} \right] \end{bmatrix} \in \mathbb{R}^{\sum_{k=1}^N r_{\text{out},k}^{(l)}}. \tag{12}$$

Then $\boldsymbol{z}_{kj}^{(l+1)}$ in the $(l+1)$-th hidden layer can be expressed by the following recursive form

$$\begin{aligned} \boldsymbol{z}_{kj}^{(l+1)} &= \left( \boldsymbol{I}_{kj}^{(l+1)} \right)^\top \left\{ \boldsymbol{\beta}^{(l)} + \boldsymbol{O}^{(l)} \boldsymbol{h} \left( \boldsymbol{z}_{11}^{(l)}, \ldots, \boldsymbol{z}_{NN}^{(l)} \right) \right\} \\ &= \left( \boldsymbol{I}_{kj}^{(l+1)} \right)^\top \boldsymbol{\beta}^{(l)} + \left( \boldsymbol{I}_{kj}^{(l+1)} \right)^\top \boldsymbol{O}^{(l)} \boldsymbol{h} \left( \boldsymbol{z}_{11}^{(l)}, \ldots, \boldsymbol{z}_{NN}^{(l)} \right), \end{aligned} \tag{13}$$

where $k = 1, \ldots, N$, and $j = 1, \ldots, k$.

In order to reduce some intermediate parameters in DNNs, we introduce new lower-dimensional matrices (vectors) to replace the product of some matrices in the above Eq. 13. Namely, $\boldsymbol{v}_{kj}^{(l)} = \left( \boldsymbol{I}_{kj}^{(l+1)} \right)^\top \boldsymbol{\beta}^{(l)}$ and $\boldsymbol{T}_{kj}^{(l)} = \left( \boldsymbol{I}_{kj}^{(l+1)} \right)^\top \boldsymbol{O}^{(l)}$. By doing so, we can simplify Eq. 13 as

$$\boldsymbol{z}_{kj}^{(l+1)} = \boldsymbol{v}_{kj}^{(l)} + \boldsymbol{T}_{kj}^{(l)} \boldsymbol{h} \left( \boldsymbol{z}_{11}^{(l)}, \ldots, \boldsymbol{z}_{NN}^{(l)} \right), \quad k = 1, \ldots, N; \quad j = 1, \ldots, k \tag{14}$$

where $\boldsymbol{v}_{kj}^{(l)} \in \mathbb{R}^{r_{\text{in},k,j}^{(l+1)}}$ and $\boldsymbol{T}_{kj}^{(l)} \in \mathbb{R}^{r_{\text{in},k,j}^{(l+1)} \times \sum_{k=1}^N r_{\text{out},k}^{(l)}}$ are the *new parameters* in DNNs.

Finally, we can use Eq. 14 above to implement a $L$-layer reducible TaylorNet. The feedforward method of a single layer is summarized in Algorithm 1 in Appendix C.

**Remark 4.2.** *In R-TaylorNet, the computational complexity of calculating hidden features* $z_{11}^{(l+1)}, \ldots, z_{NN}^{(l+1)}$ *from* $z_{11}^{(l)}, \ldots, z_{NN}^{(l)}$ *in the l-th layer can be reduced by* $\mathcal{O}(\sum_{k=1}^{N}(d^{(l+1)}r_{\text{out},k}^{(l)} + d^{(l+1)}\sum_{j=1}^{k} r_{\text{in},k,j}^{(l+1)} - (\sum_{m=1}^{N} r_{\text{out},m}^{(l)})(\sum_{j=1}^{k} r_{\text{in},k,j}^{(l+1)})))$ *compared to the original TaylorNet.*

**Taylor-Mixer**. Building on the R-TaylorNet, we also propose a new Taylor-Mixer that replaces both the MLP layers and non-linear activation functions in the MLP-Mixer Tolstikhin et al. (2021) with Taylor layer. The resulting Taylor-Mixer can be applied to image classification and natural language processing (NLP).

## 4.3 TAYLOR INITIALIZATION

We also develop a robust Taylor initialization method to mitigate the training instability caused by higher-order terms. For simplicity, we omit superscript $(l)$ unless otherwise specified. Following Xavier Glorot & Bengio (2010) and Kaiming initialization He et al. (2015), we assume that: 1) the input elements (variables) of the $l$-th layer, denoted by $y^{(l-1)}$, follow independent zero-mean Gaussian distribution. 2) the weights in $O_k, G_k, I_{kj}$ ($k = 1, \ldots, N$; $j = 1, \ldots, k$) are initialized independently with zero mean. 3) $\beta$ is initialized to 0. Then we have the following proposition.

**Proposition 4.1.** *The variance of input and output variables of the l-th layer satisfies:*

$$\left(\sigma_y^{(l)}\right)^2 = \sum_{k=1}^{N} \left(r_{\text{out},k}\sigma_{O,k}^2\right) \left(\prod_{j=1}^{k} r_{\text{in},k,j}\sigma_{G,k}^2\right) \left(\frac{(d+2k-2)!!}{(d-2)!!}\sigma_{I,k}^{2k}\right) \left(\sigma_y^{(l-1)}\right)^{2k} \quad (15)$$

where !! denotes double factorial, $\left(\sigma_y^{(l)}\right)^2$ and $\left(\sigma_y^{(l-1)}\right)^2$ denote the variance of $y^{(l)}$ and $y^{(l-1)}$. And $\sigma_{O,k}^2, \sigma_{G,k}^2$, and $\sigma_{I,k}^2$ denote the variance of the weights in $O_k$, $G_k$, and $I_{kj}$ ($k = 1, \ldots, N$; $j = 1, \ldots, k$), respectively.

Following the prior works He et al. (2015); Glorot & Bengio (2010), we should enforce $\left(\sigma_y^{(l)}\right)^2 = \left(\sigma_y^{(l-1)}\right)^2$ for stabilizing the model training. Plus, we would also like to ensure that all *intermediate* features inside the Taylor layer have similar variance as $\left(\sigma_y^{(l-1)}\right)^2$. To satisfy these requirements, the variance of weights should be initialized to:

$$\sigma_{O,k}^2 = \lambda_k \frac{1}{r_{\text{out},k}}, \ \sigma_{G,k}^2 = \frac{1}{\prod_{j=1}^{k} r_{\text{in},k,j}}, \ \sigma_{I,k}^{2k} = \frac{(d-2)!!}{(d+2k-2)!!}$$

$$s.t. \sum_{k=1}^{N} \lambda_k = 1 \quad (16)$$

where $\lambda_k$ is a coefficient that can be used to scale the importance of the $k$-th-order term. Please refer to the theoretical analysis of Proposition 4.1 in Appendix E.

Similarly, we have also developed an initialization method for Reducible TaylorNet, please refer to Appendix F for more details.

## 4.4 CONNECTIONS TO EXISTING MODELS

We present the connection between TaylorNet and some existing neural networks. According to Eq. 3, one-layer TaylorNet of order 1 is a linear function, $f(x) = f(x_0) + J(x - x_0)$, where $J$ is the Jacobian matrix of $f(x)$ at $x = x_0$. Thus, the TaylorNet of order 1 can be viewed as fully connected layers in deep neural networks. The second order term in a Taylor layer can be expressed as $\mathcal{H}_{(1)}[(x - x_0) \otimes (x - x_0)]$, where $\mathcal{H}$ is the scaled second-order derivative tensor of $f(x)$ at $x = x_0$. The Kronecker product of $x - x_0$ can be viewed as the pixel-level attention, analogous to the token-level attention in Transformer Vaswani et al. (2017); Dosovitskiy et al. (2020). Finally, our TaylorNet adopts higher-order terms to compensate for the residual errors, as shown in Fig. 1 (b), which shares the similar philosophy as a residual (or highway) network Srivastava et al. (2015); He et al. (2016).

## 5 EXPERIMENTS

We first verify the effectiveness of the proposed Taylor initialization method. Then we evaluate the performance of the proposed methods in three different applications, including image classification, explainable dynamical systems, and NLP. The detailed model configurations and hyperparameter settings are presented in Appendix D

### 5.1 TAYLOR INITIALIZATION

First of all, we compare our Taylor initialization with two commonly used initialization approaches: Xavier and Kaiming initialization. In this task, we conduct experiments on CIFAR10 using four-layer Taylor-Mixer (34.4M parameters) as described in Section 4.2. Fig. 2 illustrates the comparison results of different methods using 3 random seeds. We can observe that our Taylor initialization can significantly increase the classification accuracy by over 10% compared to the next best approach, Xavier initialization. The primary reason why both Xavier and Kaiming initialization do not perform well is that they fail to ensure the same variance for input and output at each layer.

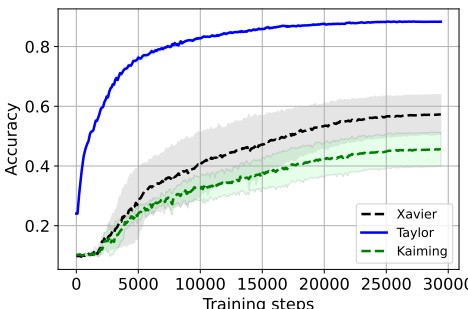

Figure 2: Accuracy comparison of different initialization methods using 3 random seeds.

### 5.2 EVALUATION ON IMAGE CLASSIFICATION

We evaluate the performance of our proposed Taylor-Mixer on image classification. We compare it with the MLP-Mixer Tolstikhin et al. (2021), which can achieve competitive results on image classification benchmarks. In our experiment, we choose the point of expansion at $x_0 = 0$ for Taylor-Mixer since the input data are normalized. Following the similar method in MLP-Mixer, we pre-train our model on ILSVRC2012 ImageNet that contains 1.3M training examples and 1k classes, and then test it on CIFAR10 and CIFAR100 datasets. Also, this work adopts the same data augmentation techniques, including RandAugment Cubuk et al. (2020), mixup Zhang et al. (2018), and stochastic depth Huang et al. (2016). In addition, we use the same fine-tuning strategy as MLP-Mixer. The detailed parameter settings are introduced in Appendix D.1. Table 1 shows the performance comparison of our Taylor-Mixer and the baselines under different model sizes. We can see that the proposed Taylor-Mixer performs better than Π-nets and achieves comparable accuracy to the MLP-Mixer with fewer model parameters on both CIFAR10 and CIFAR100. In particular, the parameters of our Base model can be reduced by about 42% compared to the MLP-Mixer. Therefore, we can conclude that our Taylor-Mixer outperforms the MLP-Mixer.

Table 1: Performance comparison for different methods using 5 random seeds. Here Small/16 and Base/32 mean the patch size is $16 \times 16$ and $32 \times 32$, respectively. We can observe that Taylor-Mixer has slightly higher accuracy but fewer parameters than the MLP-Mixer. In particular, our Base model exhibits a significant reduction in model parameters.

| Models | Small/16 | | | Base/32 | | |
|---|---|---|---|---|---|---|
| | CIFAR10 (%) | CIFAR100 (%) | Parameters (M) | CIFAR10 (%) | CIFAR100 (%) | Parameters (M) |
| MLP-Mixer | 93.21±0.08 | 74.35±0.08 | 18 | 94.16±0.16 | 76.30±0.25 | 59.6 |
| Π-nets | NA | NA | NA | 88.12±0.02 | 67.83±0.032 | 37 |
| Taylor-Mixer | **93.68±0.04** | **74.63±0.14** | **17.2** | **94.97±0.33** | **79.00±0.40** | **34.4** |

### 5.3 EVALUATION ON DYNAMICAL SYSTEMS

Next, we apply our TaylorNet to predict and interpret the dynamics of physical systems. We evaluate it on two dynamical systems, Duffing equation and High-dimensional non-linear flow attractor. To train our model, we generate 100 trajectories by randomly choosing 100 initial conditions. Then we use the 20 trajectories generated from 20 different initial conditions as the validation data. The time span of each trajectory is $t = 0, 0.01, 0.02, \ldots, 10$ with sampling time, 0.01. Thus, we can

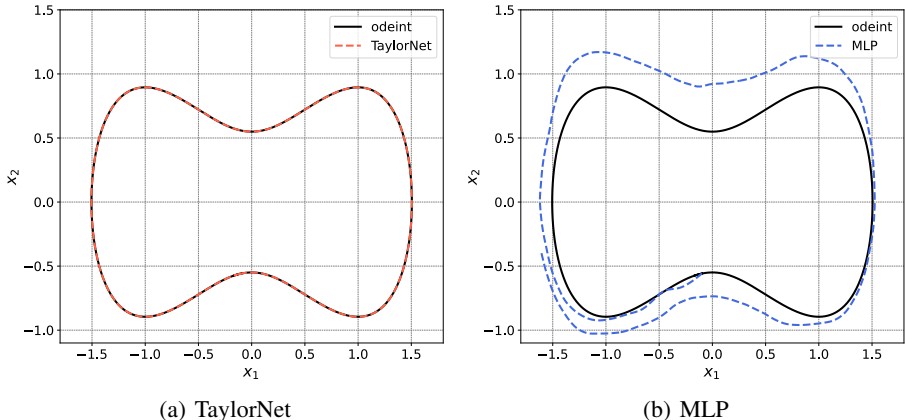

(a) TaylorNet

(b) MLP

Figure 3: Trajectory prediction of different methods on Duffing Equation.

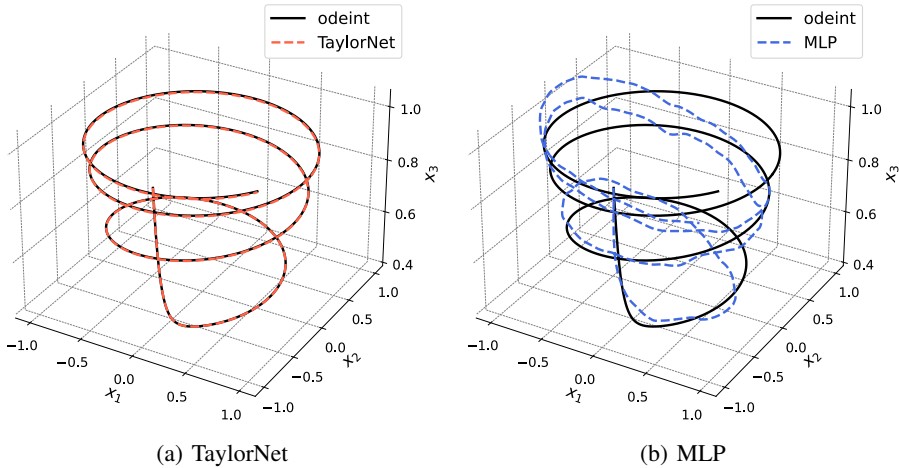

(a) TaylorNet

(b) MLP

Figure 4: Trajectory prediction of different methods on Non-linear Fluid Flow.

convert a continuous dynamical system into a discrete dynamical system, $x_{k+1} = f(x_k)$. Next, we use regression technique to predict the next state using TaylorNet. We compare our approach with two methods: ODE solver (ground truth), called odeint, from SciPy package and 3-layer MLP.

**Duffing equation.** We first adopt TaylorNet to predict the dynamics of Duffing equation, given by

$$\ddot{x} = x - x^3 \implies \begin{cases} \dot{x}_1 = x_2 \\ \dot{x}_2 = x_1 - x_1^3 \end{cases}. \tag{17}$$

In our experiment, we choose $x_1(0), x_2(0) \in [-1, 1]$.

Fig. 3 illustrates the trajectory prediction of different methods on Duffing dynamics using *one random initial condition*. We can observe from it that our TaylorNet can attain very good trajectory prediction and its error is $1.492 \times 10^{-7}$ compared to the ODE solver, odeint, from SciPy package. It thus significantly outperforms the MLP whose error is about $0.3514$. More importantly, since our method does not use activation functions, it has the ability to explicitly learn the dynamical systems in the following Eq. 18. After comparing to the original Duffing equation, we can see that our predicted model is very close to the ground-truth model in Eq.17.

$$\begin{aligned} \dot{x}_1 &\approx 1.001x_2, \\ \dot{x}_2 &\approx 1.001x_1 - 1.001x_1^3. \end{aligned} \tag{18}$$

**High-dimensional non-linear flow attractor.** We then apply our method to predict the dynamics of non-linear fluid flow. According to Noack et al. (2003), the dynamical system can be described by

the following low-dimensional model.

$$\dot{x}_1 = \mu x_1 - \omega x_2 + A x_1 x_3,$$
$$\dot{x}_2 = \omega x_1 + \mu x_2 + A x_2 x_3, \qquad (19)$$
$$\dot{x}_3 = -\lambda(x_3 - x_1^2 - x_2^2).$$

Following the prior work Lusch et al. (2018), we choose $\mu = 0.1$, $\omega = 1$, $\lambda = 10$, $A = -0.1$, and $x_1(0), x_2(0) \in [-1.1, 1.1]$, $x_3(0) \in [0, 2.42]$ in the experiment.

Fig. 4 shows the trajectory predictions of flow attractor using different methods. We can observe that our approach can accurately predict the trajectory as the ODE solver, odeint (ground truth). In addition, the error of our method is about $3.361 \times 10^{-6}$, which is three orders of magnitude smaller than that of the MLP ($4.447 \times 10^{-3}$). Finally, we leverage our TaylorNet to reconstruct the dynamical system in the following. We can see the predicted model is very close to the ground truth in the above Eq. 19. Based on these two examples, we can conclude that our TaylorNet is able to explicitly learn and interpret the dynamics of some physical systems with polynomials.

$$\dot{x}_1 \approx 0.095 x_1 - 1.003 x_2 - 0.100 x_1 x_2,$$
$$\dot{x}_2 \approx 1.002 x_1 + 0.095 x_2 + 0.100 x_2 x_3, \qquad (20)$$
$$\dot{x}_3 \approx -9.513 x_3 + 9.521 x_1^2 + 9.521 x_2^2.$$

### 5.4 Evaluation on NLP

Finally, we also explore our method in NLP applications. In this work, we use sentiment analysis on IMDB Maas et al. (2011) as a running example. Similar to image classification, we propose a new Taylor-NLP that replaces all the MLP layers, other than those in the attention mechanism, in the recently developed pNLP-Mixer Fusco et al. (2022). The parameter settings are described in Appendix D.3. Table 2 illustrates the performance comparison of different methods, and our Taylor-NLP attains comparable accuracy to the pNLP-Mixer XL but with much fewer parameters. Importantly, our Taylor-NLP does not use activation functions.

Table 2: Performance comparison of the proposed Taylor-NLP and pNLP-Mixer using IMDB dataset. Our results are averaged from 3 random seeds

| Model | Accuracy | F1 | Parameters |
|---|---|---|---|
| pNLP-Mixer XS | 80.95 | 80.95 | 1.2M |
| pNLP-Mixer Base | 81.46 | 81.38 | 2.0M |
| pNLP-Mixer XL | 82.15 | 82.66 | 4.9M |
| Taylor-NLP | **82.98$\pm$ 0.17** | **82.88$\pm$ 0.26** | **908K** |

## 6 Conclusion

This paper developed a Taylor-driven generic neural architecture, called TaylorNet, that is able to naturally introduce inductive bias to deep neural networks (DNNs). Different from classical DNNs, our TaylorNet adopted higher-order terms to replace the conventional non-linear activation functions. More specifically, we first proposed a lightweight Taylor Neural Network (TaylorNet) based on Tucker decomposition. As an extension, we also developed a reducible TaylorNet that can remove redundant parameters in hidden layers to improve computational efficiency. Then we proposed a new Taylor-Mixer that replaces both the MLP layers and activation functions in the MLP-Mixer with Taylor layers. In order to improve the model performance, a novel Taylor initialization approach was proposed. Evaluation results illustrated that the proposed method can achieve comparable accuracy to the baselines on image classification and sentiment analysis in NLP. In particular, our approach can significantly reduce the number of desired model parameters on image classification. Importantly, our approach could explicitly learn and interpret some dynamical systems with polynomials, making way for explainable ML.

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

# A    NOTATIONS

We summarize the main notations used throughout the paper in the following table.

Table 3: Summary of notations.

| Notation | Dimension(s) | Definition |
|---|---|---|
| $\otimes, \times_n, \bar{\times}_n$ | - | Kronecker product, mode-$n$ matrix product, mode-$n$ vector product |
| $k, N$ | $\mathbb{N}$ | term order of Taylor polynomial , total order of Taylor polynomial |
| $l$ | $\mathbb{N}$ | the layer number in Reducible TaylorNet |
| $r_{\mathrm{in},k,j}$ | $\mathbb{N}$ | Tucker ranks corresponds to the $j$-th input dimension in the $k$-th order in TaylorNet |
| $r_{\mathrm{out},k}$ | $\mathbb{N}$ | Tucker ranks corresponds to the output dimension in the $k$-th order in TaylorNet |
| $\Delta\boldsymbol{x}$ | $\mathbb{R}^d$ | Input of TaylorNet/Reducible TaylorNet |
| $\boldsymbol{y}$ or $\boldsymbol{f}(\boldsymbol{x})$ | $\mathbb{R}^o$ | Output of TaylorNet/Reducible TaylorNet |
| $\boldsymbol{z}_{kj}^{(l)} = \left(\boldsymbol{I}_{kj}^{(l)}\right)^{\top} \boldsymbol{y}^{(l-1)}$ | $\mathbb{R}^{r_{\mathrm{in},k,j}^{(l)}}$ | Pre-G hidden features of $l$-th layer in TaylorNet |
| $\boldsymbol{h}\left(\boldsymbol{z}_{11}^{(l)}, \ldots, \boldsymbol{z}_{NN}^{(l)}\right)$ | $\mathbb{R}^{\sum_{k=1}^{N} r_{\mathrm{out},k}^{(l)}}$ | Post-G hidden features of $l$-th layer in TaylorNet |
| $\boldsymbol{\beta} = \boldsymbol{f}(\boldsymbol{x}_0)$ | $\mathbb{R}^o$ | Learnable vector parameter |
| $\boldsymbol{\mathcal{G}}^{[k]}$ | $\mathbb{R}^{r_{\mathrm{out},k} \prod_{j=1}^{k} \times r_{\mathrm{in},k,j}}$ | Learnable core tensor of TaylorNet |
| $\boldsymbol{G} \overset{\mathrm{def}}{=} \boldsymbol{\mathcal{G}}_{(1)}^{[k]}$ | $\mathbb{R}^{r_{\mathrm{out},k}^{[k]} \times \prod_{j=1}^{k} r_{\mathrm{in},k,j}}$ | mode-1 matricization of $\boldsymbol{\mathcal{G}}^{[k]}$ |
| $\boldsymbol{I}_{kj}$ | $\mathbb{R}^{d \times r_{\mathrm{in},k,j}}$ | Learnable input factor matrices of TaylorNet |
| $\boldsymbol{O}_k$ | $\mathbb{R}^{o \times r_{\mathrm{out},k}}$ | Learnable output factor matrices of TaylorNet |
| $\boldsymbol{v}_{kj}^{(l)}$ | $\mathbb{R}^{r_{\mathrm{in},k,j}^{(l)}}$ | New learnable vector parameters in Reducible TaylorNet |
| $\boldsymbol{T}_{kj}^{(l)}$ | $\mathbb{R}^{r_{\mathrm{in},k,j}^{(l)} \times \sum_{k=1}^{N} r_{\mathrm{out},k}^{(l-1)}}$ | New learnable matrix parameters in Reducible TaylorNet |
| $\left(\sigma_y^{(l)}\right)^2, \left(\sigma_y^{(l-1)}\right)^2, \sigma_{O,k}^2, \sigma_{G,k}^2, \sigma_{I,k}^2$ | $\mathbb{N}$ | Initialization variance for $\boldsymbol{y}^{(l)}, \boldsymbol{y}^{(l-1)}, \boldsymbol{O}_k^{(l)}, \boldsymbol{G}_k^{(l)}, \boldsymbol{I}_k^{(l)}$ |
| $\lambda_k$ | $\mathbb{N}$ | Initialization coefficient for $\sigma_{O,k}^2$ |

## B PROPERTIES OF TENSOR MODE-$n$ PRODUCT

**Lemma B.1.** *For mode-$n$ matrix product, it satisfies the commutative law (Kolda & Bader, 2009)*

$$\boldsymbol{\mathcal{X}} \times_m \boldsymbol{A} \times_n \boldsymbol{B} = \boldsymbol{\mathcal{X}} \times_n \boldsymbol{B} \times_m \boldsymbol{A}, \tag{21}$$

*which means that the order of multiplication is irrelevant when it comes to different modes in a series of mode matrix product.*

**Lemma B.2.** *For mode-$n$ matrix product, it satisfies the following associative property(Kolda & Bader, 2009)*

$$\boldsymbol{\mathcal{X}} \times_n \boldsymbol{A} \times_n \boldsymbol{B} = \boldsymbol{\mathcal{X}} \times_n (\boldsymbol{BA}). \tag{22}$$

*Proof.* Based on Lemma. B.1, the $k$-th term of Taylor expansion in Eq. 6 can be rewritten as

$$\boldsymbol{\mathcal{W}}^{[k]} \prod_{j=2}^{k+1} \bar{\times}_j \Delta \boldsymbol{x} = \boldsymbol{\mathcal{G}}^{[k]} \times_1 \boldsymbol{O}_k \left( \prod_{j=1}^{k} \times_{j+1} \boldsymbol{I}_{kj} \times_{j+1} \Delta \boldsymbol{x}^\top \right) \tag{23}$$

Based on Lemma B.2, the above Eq. 23 can be reformulated as

$$\boldsymbol{\mathcal{W}}^{[k]} \prod_{j=2}^{k+1} \bar{\times}_j \Delta \boldsymbol{x} = \boldsymbol{\mathcal{G}}^{[k]} \times_1 \boldsymbol{O}_k \left[ \prod_{j=1}^{k} \times_{j+1} \left( \Delta \boldsymbol{x}^\top \boldsymbol{I}_{kj} \right) \right] \in \mathbb{R}^o. \tag{24}$$

$\square$

Proof finished.

## C FEEDFORWARD METHOD FOR REDUCIBLE TAYLORNET

We summarize the feedforward method for Reducible TaylorNet below.

---

**Algorithm 1:** Feedforward Method for Reducible TaylorNet

---

**Input** : $\Delta \boldsymbol{x} \in \mathbb{R}^d$
**Output** : $\boldsymbol{y} \in \mathbb{R}^o$
Initialize $\boldsymbol{v}_{kj}^{(l)}, \boldsymbol{T}_{kj}^{(l)}, \boldsymbol{\beta}^{(L)}, \boldsymbol{O}^{(L)}, \boldsymbol{G}_k^{(l)}, \boldsymbol{I}_{kj}^{(1)}$
/\*from the 1-st layer to the $L$-th layer                    \*/
**for** $l = 1, \ldots, L$ **do**
    /\*from the 1-st order to the $N$-th order            \*/
    **for** $k = 1, \ldots, N$ **do**
        **for** $j = 1, \ldots, k$ **do**
            **if** $l = 1$ **then**
                $\boldsymbol{z}_{kj}^{(1)} = \left( \boldsymbol{I}_{kj}^{(1)} \right)^\top \Delta \boldsymbol{x}$
            **else**
                $\boldsymbol{z}_{kj}^{(l)} = \boldsymbol{v}_{kj}^{(l-1)} + \boldsymbol{T}_{kj}^{(l-1)} \boldsymbol{h} \left( \boldsymbol{z}_{11}^{(l-1)}, \ldots, \boldsymbol{z}_{NN}^{(l-1)} \right)$
            **end**
        **end**
    **end**
**end**
$\boldsymbol{y} = \boldsymbol{y}^{(L)} = \boldsymbol{\beta}^{(L)} + \boldsymbol{O}^{(L)} \boldsymbol{h} \left( \boldsymbol{z}_{11}^{(L)}, \ldots, \boldsymbol{z}_{NN}^{(L)} \right)$

---

## D MODEL CONFIGURATIONS AND PARAMETER SETTINGS

In this section, we present the detailed model configurations and parameter settings for the following four different tasks.

## D.1 EXPERIMENTAL DETAILS FOR IMAGE CLASSIFICATION

For image classification, our Taylor-Mixer is built on the existing MLP-mixer Tolstikhin et al. (2021). Thus we follow the experimental settings in the MLP-mixer for pre-training and fine-tuning, unless stated otherwise.

**Pre-training**. We pre-train all the models at resolution 224 using linear learning rate warm-up and cosine learning rate decay. We set the batch-size to 1024 for Base and Small model due to GPU memory capacity limitation in our servers. Since the input data are normalized, we choose Taylor expansion point $x_0 = 0$ in our model. The detailed model configurations and parameters settings are presented in Table 4. Our Taylor-Mixer is set to 2 and 4 layers in our experiments. The Tucker rank of input and output factor matrices are set to 110 and 140, respectively. We describe the rule of thumb for choosing the ranks as follows. For a $N$-th order Taylor layer with input and output rank $r_{\text{in},k,j}$ and $r_{\text{out},k}$, the effective width of this layer is approximately $\min(\sum_{k=1}^{N} \sum_{j=1}^{k} j r_{\text{in},k,j}, \sum_{k=1}^{N} r_{\text{out},k})$. Therefore, in order to achieve larger width with fixed number of parameters, we should set $\sum_{k=1}^{N} \sum_{j=1}^{k} j r_{\text{in},k,j} \approx \sum_{k=1}^{N} r_{\text{out},k}$. Then we can scale $r_{\text{in},k,j}$ and $r_{\text{out},k}$ together to adjust the number of parameters of the model.

Table 4: Configurations of Taylor-Mixer architectures for different model scales: Small and Base.

| Specification | Small/16 | Base/32 |
|---|---|---|
| Number of layers | 2 | 4 |
| Rank of input factor matrices | 110 | 110 |
| Rank of output factor matrices | 140 | 140 |
| Patch resolution $P \times P$ | $16 \times 16$ | $32 \times 32$ |
| Hidden size | 512 | 768 |
| Sequence length $S$ | 196 | 49 |
| Dimension for channel-mixing $D_c$ | 2048 | 3072 |
| Dimension for token-mixing $D_s$ | 256 | 384 |
| Parameters (M) | 17.2 | 34.4 |
| Initialization $\lambda_1, \lambda_2$ | 0.99, 0.01 | 0.99, 0.01 |

**Fine-tuning**. For a fair comparison, we follow the experimental settings in the MLP-Mixer work. We use momentum SGD optimizer and a cosine learning rate scheduler with a linear warm-up. The batch size of fine-tuning is set to 512. We also use gradient clipping at global norm 1. In addition, we do not use dropout, the same as MLP-Mixer.

## D.2 EXPERIMENTAL DETAILS FOR DYNAMICAL SYSTEMS

In this experiment, we leverage one-layer TaylorNet based on Tucker decomposition. The rank of each dimension in the core tensor is set to 16. In addition, the batch size is set to 128. We use Adam optimizer with learning rate 0.001.

## D.3 EXPERIMENTAL DETAILS FOR NLP

For sentiment analysis in NLP, we follow the same experimental setup in the pNLP-Mixer Fusco et al. (2022) unless otherwise stated. Following pNLP-Mixer, we set the batch size and hidden size to 256 and 256 respectively. We use Adam optimizer with learning rate $10^{-4}$. Different from pNLP-Mixer, the length of input tokens is set to 512. we use BERT embeddings for a token by averaging the embeddings of its subword units. In order to make the number of parameters similar to that of pNLP-Mixer, we choose 2-layers Taylor-NLP with the rank of 30 and 50 for the input and output matrices respectively. We also use dropout of 0.5 and weight decay of 0.01 to mitigate overfit problem. Initialization $\lambda_1, \lambda_2$ are set to 0.99, 0.01.

# E    ANALYSIS ON TAYLOR INITIALIZATION IN PROPOSITION 4.1

In this section, we offer the theoretical analysis on Taylor initialization in Proposition 4.1. First, we introduce two lemmas used to decompose the variance of the output variables.

**Lemma E.1.** *Suppose that (i) $w_1$ is independent to $w_2, x_1$, and $x_2$, (ii) $w_2$ is independent to $w_1, x_1$ and $x_2$, and (iii) $\mathbb{E}[w_1] = \mathbb{E}[w_2] = 0$, then we have*

$$\mathrm{Cov}[w_1 x_1, w_2 x_2] = 0 \tag{25}$$

*Proof.*

$$\begin{aligned}
\mathrm{Cov}[w_1 x_1, w_2 x_2] &= \mathbb{E}[w_1 x_1 w_2 x_2] - \mathbb{E}[w_1 x_1]\mathbb{E}[w_2 x_2] \\
&= \mathbb{E}[w_1]\mathbb{E}[w_2]\mathbb{E}[x_1 x_2] - \mathbb{E}[w_1]\mathbb{E}[x_1]\mathbb{E}[w_2]\mathbb{E}[x_2] \\
&= 0,
\end{aligned} \tag{26}$$

completing the proof.    □

**Lemma E.2.** *If $\mathbb{E}[w_j] = 0$, and $w_j$ is independent to $w_k$ and $x_i$, for all $j, k \neq j, i$, then we have*

$$\mathrm{Var}\Big[\sum_i w_i x_i\Big] = \sum_i \mathrm{Var}[w_i]\mathbb{E}[x_i^2] \tag{27}$$

*Proof.*

$$\mathrm{Var}\Big[\sum_i w_i x_i\Big] = \sum_i \mathrm{Var}[w_i x_i] + \sum_i \sum_{j>i} 2\mathrm{Cov}[w_i x_i, w_j x_j] \tag{28}$$

According to Lemma E.1, we can eliminate the second term above. Thus we have

$$\begin{aligned}
\mathrm{Var}\Big[\sum_i w_i x_i\Big] &= \sum_i \mathrm{Var}[w_i x_i] \\
&= \sum_i \mathbb{E}[w_i^2 x_i^2] - \mathbb{E}^2[w_i]\mathbb{E}^2[x_i] \\
&= \sum_i \mathbb{E}[w_i^2]\mathbb{E}[x_i^2] \\
&= \sum_i \mathrm{Var}[w_i]\mathbb{E}[x_i^2],
\end{aligned} \tag{29}$$

completing the proof.    □

Next, we introduce a conjecture which will be used in our main proof.

**Conjecture E.1.** *For a random vector $\boldsymbol{x}$ following standard multivariate Gaussian distribution and for arbitrary $k \leq d$, we have*

$$\sum_{i_1=1, i_2=1, \ldots, i_k=1}^{d} \mathbb{E}[\boldsymbol{x}_{i_1}^2 \boldsymbol{x}_{i_2}^2 \ldots \boldsymbol{x}_{i_k}^2] = \frac{(d + 2k - 2)!!}{(d-2)!!}, \tag{30}$$

*where $i_1, \ldots, i_k$ denote the indices of $\boldsymbol{x}$, $d$ is the dimension of the input and $k$ is the order of the Taylor series expansion.*

We first offer a proof of this conjecture for small $k$ ($k = 1, 2, 3, 4$) using enumeration below. This is because we often choose lower-order Taylor expansion for each layer in TaylorNet considering the computational cost.

*Proof.* We use the property of Unit Gaussian distribution that

$$\mathbb{E}[\mathrm{x}^p] = (p-1)!! \tag{31}$$

where x follows Unit Gaussian distribution and $p$ is a positive even integer.

For $k = 1$, Eq. 30 can be rewritten as

$$\sum_{i_1=1}^{d} \mathbb{E}[\boldsymbol{x}_{i_1}^2] = d \times 1 = d \tag{32}$$

For $k = 2$, Eq. 30 can be rewritten and proved as

$$\sum_{i_1=1,i_2=1}^{d} \mathbb{E}[\boldsymbol{x}_{i_1}^2 \boldsymbol{x}_{i_2}^2] = \sum_{i_1=1}^{d} \mathbb{E}[\boldsymbol{x}_{i_1}^4] + \sum_{i_1,i_2 \neq i_1}^{d} \mathbb{E}[\boldsymbol{x}_{i_1}^2 \boldsymbol{x}_{i_2}^2] = d \times 3 + d(d-1) \times 1 = d(d+2) \tag{33}$$

For $k = 3$, Eq. 30 can be rewritten and proved as

$$\sum_{i_1,i_2,i_3}^{d} \mathbb{E}[\boldsymbol{x}_{i_1}^2 \boldsymbol{x}_{i_2}^2 \boldsymbol{x}_{i_3}^2]$$
$$= \sum_{i_1}^{d} \mathbb{E}[\boldsymbol{x}_{i_1}^6] + C_3^1 \sum_{i_1,i_2 \neq i_1}^{d} \mathbb{E}[\boldsymbol{x}_{i_1}^4 \boldsymbol{x}_{i_2}^2] + \sum_{i_1,i_2 \neq i_1,i_3 \neq i_1,i_2}^{d} \mathbb{E}[\boldsymbol{x}_{i_1}^2 \boldsymbol{x}_{i_2}^2 \boldsymbol{x}_{i_3}^2]$$
$$= d \times 15 + 3 \times d(d-1) \times 3 + d(d-1)(d-2) \times 1$$
$$= d(d+2)(d+4) \tag{34}$$

For $k = 4$, Eq. 30 can be rewritten and proved as

$$\sum_{i_1,i_2,i_3,i_4}^{d} \mathbb{E}[\boldsymbol{x}_{i_1}^2 \boldsymbol{x}_{i_2}^2 \boldsymbol{x}_{i_3}^2 \boldsymbol{x}_{i_4}^2]$$
$$= \sum_{i_1}^{d} \mathbb{E}[\boldsymbol{x}_{i_1}^8] + C_4^1 \sum_{i_1,i_2 \neq i_1}^{d} \mathbb{E}[\boldsymbol{x}_{i_1}^6 \boldsymbol{x}_{i_2}^2] + \frac{C_4^2}{2} \sum_{i_1,i_2 \neq i_1}^{d} \mathbb{E}[\boldsymbol{x}_{i_1}^4 \boldsymbol{x}_{i_2}^4] +$$
$$C_4^2 \sum_{i_1,i_2 \neq i_1,i_3 \neq i_1,i_2}^{d} \mathbb{E}[\boldsymbol{x}_{i_1}^4 \boldsymbol{x}_{i_2}^2 \boldsymbol{x}_{i_3}^2] + \sum_{i_1,i_2 \neq i_1,i_3 \neq i_1,i_2,i_4 \neq i_1,i_2,i_3}^{d} \mathbb{E}[\boldsymbol{x}_{i_1}^2 \boldsymbol{x}_{i_2}^2 \boldsymbol{x}_{i_3}^2 \boldsymbol{x}_{i_4}^2]$$
$$= d \times 105 + 4 \times d(d-1) \times 15 + 3 \times d(d-1) \times 9 +$$
$$6 \times d(d-1)(d-2) \times 3 + d(d-1)(d-2)(d-3) \times 1$$
$$= d(d+2)(d+4)(d+6), \tag{35}$$
completing the proof. $\square$

Based on the above proof for small $k$, we can extrapolate Conjecture E.1 to all $k \leq d$. We have empirically validated that the this conjecture still holds for $d \in 1, \ldots, 64, k \leq d$ using computer simulation. Nevertheless, we are still attempting to prove it thoroughly for our future work.

Based on the above lemmas and Conjecture E.1, we can offer the proof of Proposition 4.1 below.

*Proof.* We first define two hidden features in the Taylor layer $\tilde{\boldsymbol{z}}_k = \left( \boldsymbol{I}_{kk}^\top \boldsymbol{y}^{(l-1)} \right) \otimes \cdots \otimes \left( \boldsymbol{I}_{k1}^\top \boldsymbol{y}^{(l-1)} \right)$, and $\boldsymbol{h}_k = \boldsymbol{G}_k \tilde{\boldsymbol{z}}_k$. Let $\sigma_{\boldsymbol{h},k}^2 = \text{Var}[(\boldsymbol{h}_k)_j]$ denotes the variance of $\boldsymbol{h}_k$, and $\nu_{z,k} = \mathbb{E}[(\tilde{\boldsymbol{z}}_k)_i^2]$ denotes the second order origin moment of $\tilde{\boldsymbol{z}}_k$.

We can first derive the relationship between the variance of $\boldsymbol{y}^{(l)}$ and $\boldsymbol{h}_k$. According to Eq. 10, in TaylorNet, we have $\boldsymbol{y}^{(l)} = \boldsymbol{\beta} + \sum_{k=1}^{N} \boldsymbol{O}_k \boldsymbol{h}_k$. And it can be decomposed into

$$\boldsymbol{y}^{(l)}{}_i = \boldsymbol{\beta}_i + \sum_{k=1}^{N} \sum_{j=1}^{r_{\text{out},k}} (\boldsymbol{O}_k)_{i,j} (\boldsymbol{h}_k)_j \tag{36}$$

Therefore, according to Lemma E.2, we can derive the variance of $\boldsymbol{y}^{(l)}$ as

$$\left(\sigma_y^{(l)}\right)^2 = \sum_{k=1}^{N} \sum_{j=1}^{r_{\text{out},k}} \sigma_{O,k}^2 \mathbb{E}[(\boldsymbol{h}_k)_j^2] \tag{37}$$

Given that $\mathbb{E}[(\boldsymbol{h}_k)_j] = \mathbb{E}[\boldsymbol{G}_k \tilde{\boldsymbol{z}}_k] = 0$, the above Eq. 37 can be further simplified as

$$\left(\sigma_y^{(l)}\right)^2 = \sum_{k=1}^{N} r_{\text{out},k} \sigma_{O,k}^2 \sigma_{\boldsymbol{h},k}^2 \tag{38}$$

Next, we establish the relationship between $\sigma_{\boldsymbol{h},k}^2$ and $\nu_{z,k}$. We can decompose $\boldsymbol{h}_k = \boldsymbol{G}_k \tilde{\boldsymbol{z}}_k$ into the following formula.

$$(\boldsymbol{h}_k)_j = \sum_{i_1}^{r_{in,k,1}} \cdots \sum_{i_k}^{r_{in,k,k}} (\boldsymbol{G}_k)_{j,i_1 \times \cdots \times i_k} (\tilde{\boldsymbol{z}}_k)_{i_1 \times \cdots \times i_k} \tag{39}$$

Therefore, according to Lemma E.2, we can derive $\sigma_{\boldsymbol{h},k}^2$ as

$$\sigma_{\boldsymbol{h},k}^2 = \prod_{j=1}^{k} r_{\text{in},k,j} \sigma_{G,k}^2 \nu_{z,k} \tag{40}$$

Next, we establish the relationship between $\nu_{z,k}$ and $\left(\sigma_y^{(l-1)}\right)^2$. We can decompose $\tilde{\boldsymbol{z}}_k = \left(\boldsymbol{I}_{kk}^{\top} \boldsymbol{y}^{(l-1)}\right) \otimes \cdots \otimes \left(\boldsymbol{I}_{k1}^{\top} \boldsymbol{y}^{(l-1)}\right)$ into

$$(\tilde{\boldsymbol{z}}_k)_{i_1 \times \cdots \times i_k} = \left(\sum_{j_k}^{d} (\boldsymbol{I}_{kk})_{j_k,i_k} (\boldsymbol{y}^{(l-1)})_{j_k}\right) \cdots \left(\sum_{j_1}^{d} (\boldsymbol{I}_{k1})_{j_1,i_1} (\boldsymbol{y}^{(l-1)})_{j_1}\right) \tag{41}$$

Therefore we can derive $\nu_{z,k}$ as

$$\nu_{z,k} = \mathbb{E}[(\tilde{\boldsymbol{z}}_k)_{i_1 \times \cdots \times i_k}^2] \tag{42}$$

$$= \mathbb{E}\left[\left(\sum_{j_k}^{d} (\boldsymbol{I}_{kk})_{j_k,i_k} (\boldsymbol{y}^{(l-1)})_{j_k}\right)^2 \cdots \left(\sum_{j_1}^{d} (\boldsymbol{I}_{k1})_{j_1,i_1} (\boldsymbol{y}^{(l-1)})_{j_1}\right)^2\right]$$

$$= \mathbb{E}\left[\sum_{j_k}^{d} (\boldsymbol{I}_{kk})_{j_k,i_k}^2 (\boldsymbol{y}^{(l-1)})_{j_k}^2 \cdots \sum_{j_1}^{d} (\boldsymbol{I}_{k1})_{j_1,i_1}^2 (\boldsymbol{y}^{(l-1)})_{j_1}^2\right]$$

$$= \sum_{j_1}^{d} \cdots \sum_{j_k}^{d} \mathbb{E}\left[(\boldsymbol{I}_{kk})_{j_k,i_k}^2 \cdots (\boldsymbol{I}_{k1})_{j_1,i_1}^2\right] \mathbb{E}\left[(\boldsymbol{y}^{(l-1)})_{j_k}^2 \cdots (\boldsymbol{y}^{(l-1)})_{j_1}^2\right]$$

$$= \sum_{j_1}^{d} \cdots \sum_{j_k}^{d} \sigma_{I,k}^{2k} \mathbb{E}\left[(\boldsymbol{y}^{(l-1)})_{j_k}^2 \cdots (\boldsymbol{y}^{(l-1)})_{j_1}^2\right]$$

$$= \sigma_{I,k}^{2k} \sum_{j_1}^{d} \cdots \sum_{j_k}^{d} \mathbb{E}\left[(\boldsymbol{y}^{(l-1)})_{j_k}^2 \cdots (\boldsymbol{y}^{(l-1)})_{j_1}^2\right] \tag{43}$$

According to Conjecture E.1, we can further simplify the above Eq. 42 as

$$\nu_{z,k} = \sigma_{I,k}^{2k} \frac{(d+2k-2)!!}{(d-2)!!} (\sigma_y^{(l-1)})^{2k} \tag{44}$$

Combining Equation 38, 40 and 44, we have

$$\left(\sigma_y^{(l)}\right)^2 = \sum_{k=1}^{N} \left(r_{\text{out},k} \sigma_{O,k}^2\right) \left(\prod_{j=1}^{k} r_{\text{in},k,j} \sigma_{G,k}^2\right) \left(\frac{(d+2k-2)!!}{(d-2)!!} \sigma_{I,k}^{2k}\right) (\sigma_y^{(l-1)})^{2k} \tag{45}$$

completing the proof. $\qquad\square$

**Initialization.** In order to stabilize the model training of our TaylorNet, we should choose $\sigma_{O,k}^2, \sigma_{G,k}^2, \sigma_{I,k}^2$ such that $\left(\sigma_y^{(l)}\right)^2 = \left(\sigma_y^{(l-1)}\right)^2$. Namely, any combinations of $\sigma_{O,k}^2, \sigma_{G,k}^2, \sigma_{I,k}^2$ satisfying the following equation is a viable choice for initialization.

$$\sum_{k=1}^{N} \left(r_{\text{out},k}\sigma_{O,k}^2\right) \left(\prod_{j=1}^{k} r_{\text{in},k,j}\sigma_{G,k}^2\right) \left(\frac{(d+2k-2)!!}{(d-2)!!}\sigma_{I,k}^{2k}\right) = 1 \tag{46}$$

In addition to the requirement of $\left(\sigma_y^{(l)}\right)^2 = \left(\sigma_y^{(l-1)}\right)^2$, another desirable property is that the second order moments of intermediate features $h_k, \tilde{z}_k$ should also be equal to the variance of the input. Namely, we would also like to ensure that $\sigma_{h,k}^2 = \nu_{z,k} = \left(\sigma_y^{(l-1)}\right)^2$. Consequently, we should choose the initialization

$$\sigma_{O,k}^2 = \lambda_k \frac{1}{r_{\text{out},k}}, \ \sigma_{G,k}^2 = \frac{1}{\prod_{j=1}^{k} r_{\text{in},k,j}}, \ \sigma_{I,k}^{2k} = \frac{(d-2)!!}{(d+2k-2)!!}$$

$$s.t. \sum_{k=1}^{N} \lambda_k = 1 \tag{47}$$

## F ANALYSIS ON REDUCIBLE TAYLOR INITIALIZATION

In this section, we will elaborate the initialization method for Reducible TaylorNet (R-TaylorNet). We keep using the notations in Section 4.2, 4.3 and E. Since the original input and output variable $y^{(l-1)}, y^{(l)}$ are omitted in reduced TaylorNet, we will alternatively examine the relationship between the variance of $h^{(l-1)}$ and $h^{(l)}$. Recall that $h^{(l)}$ is defined in 4.2 and E as

$$h^{(l)} \overset{\text{def}}{=} \begin{bmatrix} h_1^{(l)} \\ h_2^{(l)} \\ \vdots \\ h_N^{(l)} \end{bmatrix} \overset{\text{def}}{=} \begin{bmatrix} G_1^{(l)} z_{11}^{(l)} \\ G_2^{(l)} \left[ z_{22}^{(l)} \otimes z_{21}^{(l)} \right] \\ \vdots \\ G_N^{(l)} \left[ z_{NN}^{(l)} \otimes \cdots \otimes z_{N1}^{(l)} \right] \end{bmatrix} \tag{48}$$

Similar to the analysis in Section 4.3, we assume that 1) all elements in $h^{(l-1)}$ follow independent zero-mean Gaussian distribution, 2) the weights in $T_k, G_k$ are initialized independently with zero mean. 3) $v_k$ is initialized to 0.

First, we can establish the relationship between $(\sigma_{h,k}^{(l)})^2$ and $\nu_{z,k}$ in the same way as described in Section E, which can be written as

$$(\sigma_{h,k}^{(l)})^2 = \prod_{j=1}^{k} r_{\text{in},k,j}\sigma_{G,k}^2 \nu_{z,k} \tag{49}$$

Next, we establish the relationship between $\nu_{z,k}$ and $(\sigma_h^{(l-1)})^2$. In reducible TaylorNet, $\tilde{z}_k$ is calculated as

$$z_{kj} = v_{kj} + T_{kj}h^{(l-1)}$$
$$\tilde{z}_k = z_{k1} \otimes \cdots \otimes z_{kk} \tag{50}$$

Below, we introduce a more fine-grained block multiplication notation of $T_{kj}h^{(l-1)}$

$$T_{kj}h^{(l-1)} = \begin{bmatrix} T_{kj1} & \cdots & T_{kjN} \end{bmatrix} \begin{bmatrix} h_1^{(l-1)} \\ h_2^{(l-1)} \\ \vdots \\ h_N^{(l-1)} \end{bmatrix} \tag{51}$$

Given that $\boldsymbol{v}_k$ is initialized to 0, we can decompose the above matrix multiplication into

$$(\tilde{\boldsymbol{z}}_k)_{i_1\times\cdots\times i_k} = \left(\sum_m^N \sum_{j_k}^{r_{\text{out},k}} (\boldsymbol{T}_{kkm})_{i_k,j_k}(\boldsymbol{h}_m^{(l-1)})_{j_k}\right) \cdots \left(\sum_m^N \sum_{j_1}^{r_{\text{out},1}} (\boldsymbol{T}_{k1m})_{i_1,j_1}(\boldsymbol{h}_m^{(l-1)})_{j_1}\right) \quad (52)$$

When choosing the initialization variance for the original TaylorNet as shown in Eq. 16, we can set different $\lambda_k$ to scale the importance of k-th-order term. Similarly. in Reducible TaylorNet, we would also like to scale the variance of $\boldsymbol{T}_{kjm}$ for different $m$. Namely, let $\sigma_{T,km}^2$ denote the variance of $\boldsymbol{T}_{kjm}$, and $\sigma_{T,k}^2$ be the standard variance for $\boldsymbol{T}_{kj}$, then they should satisfy $\sigma_{T,km}^2 = \lambda_m \sigma_{T,k}^2$.

Now we focus on 2-order Reducible TaylorNet. We can derive $\nu_{z,k}$ for $k = 1, 2$ as

$$\nu_{z,1} = \mathbb{E}\left[\left(\sum_{j_1}^{r_{\text{out},1}}((\boldsymbol{T}_{111})_{i_1,j_1}(\boldsymbol{h}_1^{(l-1)})_{j_1} + (\boldsymbol{T}_{112})_{i_1,j_1}(\boldsymbol{h}_2^{(l-1)})_{j_1})\right)^2\right]$$

$$= r_{\text{out},1}(\lambda_1 + \lambda_2)\sigma_{T,1}^2(\sigma_{\boldsymbol{h}}^{(l-1)})^2 \quad (53)$$

$$\nu_{z,2} = \mathbb{E}\left[\left(\sum_{j_1}^{r_{\text{out},2}}((\boldsymbol{T}_{211})_{i_1,j_1}(\boldsymbol{h}_1^{(l-1)})_{j_1} + (\boldsymbol{T}_{212})_{i_1,j_1}(\boldsymbol{h}_2^{(l-1)})_{j_1})\right)^2\right.$$

$$\left.\left(\sum_{j_2}^{r_{\text{out},2}}((\boldsymbol{T}_{221})_{i_2,j_2}(\boldsymbol{h}_1^{(l-1)})_{j_2} + (\boldsymbol{T}_{222})_{i_2,j_2}(\boldsymbol{h}_2^{(l-1)})_{j_2})\right)^2\right]$$

$$= \sigma_{T,2}^4 \mathbb{E}\left[\left(\sum_{j_1}^{r_{\text{out},2}}(\lambda_1(\boldsymbol{h}_1^{(l-1)})_{j_1}^2 + \lambda_2(\boldsymbol{h}_2^{(l-1)})_{j_1}^2)\right)\left(\sum_{j_2}^{r_{\text{out},2}}(\lambda_1(\boldsymbol{h}_1^{(l-1)})_{j_2}^2 + \lambda_2(\boldsymbol{h}_2^{(l-1)})_{j_2}^2)\right)\right]$$

$$= \sigma_{T,2}^4(\sigma_{\boldsymbol{h}}^{(l-1)})^2\left(2r_{\text{out},2}(\lambda_1^2 + \lambda_2^2) + r_{\text{out},2}^2\right) \quad (54)$$

**Initialization.** Using the same methodology in Section E, we need to choose $\sigma_{T,k}^2, \sigma_{G,k}^2$ such that $(\sigma_{\boldsymbol{h},k}^{(l)})^2 = (\sigma_{\boldsymbol{h}}^{(l-1)})^2$. On the other hand, we would also like to ensure that $\nu_{z,k} = (\sigma_{\boldsymbol{h}}^{(l-1)})^2$. Hence, according to Eq. 49, 53 and 54, 2-order Reducible TaylorNet in our paper should use the following initialization

$$\sigma_{G,k}^2 = \frac{1}{\prod_{j=1}^k r_{\text{in},k,j}}, \ \ \sigma_{T,1}^2 = \frac{1}{r_{\text{out},1}}, \ \ \sigma_{T,2}^4 = \frac{1}{\left(2r_{\text{out},2}(\lambda_1^2 + \lambda_2^2) + r_{\text{out},2}^2\right)}$$

$$s.t. \ \ \lambda_1 + \lambda_2 = 1 \quad (55)$$

