# OpenReview forum: "TaylorNet: A Taylor-Driven Generic Neural Architecture"
_ICLR.cc/2023/Conference — Submitted to ICLR 2023_

### Official Review · Reviewer_D6Yc · 2022-10-19

**Confidence:** 4
**Correctness:** 3
**Technical Novelty And Significance:** 2
**Empirical Novelty And Significance:** 3
**Recommendation:** 5

**Clarity, Quality, Novelty And Reproducibility:**

**Clarity and Quality:**

Some statements are not clearly given in the paper. One happens in the second to the last paragraph of sec 1, where the authors say, “Meanwhile, our Taylor initialization reach an accuracy that is ….higher than the Xavier and Kaiming initialization….”This is misleading! It makes readers recognize that the new initialization outperforms Xavier and Kaiming in deep learning. But it is not valid. We can see from sec 5.1 that the superior performance by the new initialization is ONLY evaluated for the proposed TaylorNet. It is a crucial precondition for the statement, but it is not clearly mentioned when summarising the contributions.

**Novelty:**

The novelty of this work is weak from the technical perspective (for reasons see the “strengths and weakness”). Still, it is good that the authors highlight the potential interpretability of polynomial networks.

**Reproducibility**

The experimental settings were introduced clearly.

**Strength And Weaknesses:**

++Strength

The new model can achieve comparable performance to some deep neural networks on many tasks.

— Weakness

The basic idea of the TaylorNet is not new. It is strongly close to the old-school polynomial network (not overviewed in the related works), which was widely studied 30 years ago. A good book on this point can be found in (Nikolaev and Hitoshi, 2006). More recently, similar ideas are discussed in the tensor community, such as (Qiu et al., 202 and Rabusseau et al., 2019). In these works, they discussed multi-linear functions as the learning model. The proposed initialization trick is also not novel. A similar idea has been discussed by Pan et al. (2022), which is also not cited in the paper.

*Nikolaev, Nikolay, and Hitoshi Iba. Adaptive learning of polynomial networks: genetic programming, backpropagation and Bayesian methods. Springer Science & Business Media, 2006.*

*Qiu, Hejia, et al. "On the Memory Mechanism of Tensor-Power Recurrent Models." International Conference on Artificial Intelligence and Statistics. PMLR, 2021.*

*Rabusseau, Guillaume, Tianyu Li, and Doina Precup. "Connecting weighted automata and recurrent neural networks through spectral learning." The 22nd International Conference on Artificial Intelligence and Statistics. PMLR, 2019.*

*Pan, Yu, et al. "A Unified Weight Initialization Paradigm for Tensorial Convolutional Neural Networks." International Conference on Machine Learning. PMLR, 2022.*

**Summary Of The Paper:**

This paper introduced a new type of neural layer, which is inspired by the well-known Taylor expansion. In particular, for each layer, the learning model is represented by a Takyor summation within two orders. To address the high dimensionality of the Hassien tensor, the low-rank Tucker decomposition is taken into account. This paper also discussed suitable ways for the initialization of the new layers. In the experiments, this paper considered various tasks, including image recognition, identification of dynamical systems, and NLP.

**Summary Of The Review:**

I agree that the polynomial learning model (such as the proposed TaylorNet) would be a good candidate as a universal approximator with potentially better interpretability than the existing deep neural networks. This paper clearly pointed out these potentials and also mentioned the main challenges, such as the dimension explosion and the stability. However, all these challenges are still not resolved well so far in this paper (I cannot believe that only using a good initialization can guarantee the stability of the model in a deep regime).

---

> ### Author Response · Authors · 2022-11-19
> **Response to Reviewer D6Yc**
>
> **Q3.1: The basic idea of the TaylorNet is not new. It is strongly close to the old-school polynomial network (not overviewed in the related works), which was widely studied 30 years ago (Nikolaev and Hitoshi, 2006)**.  \
> **A3.1:** We would like to present the big difference between our TaylorNet and existing works on polynomial neural networks [Nikolaev and Hitoshi, 2006]. The earlier work mainly adopted Group Method of Data Handling to learn partial descriptors in . It, however, fails to scale to high-dimensional data as mentioned in follow-up works[Chrysos et al. 2020]. In addition, we would like to point out the significant difference between our TaylorNet and $\Pi$-Net in three-fold. First of all, while $\Pi$-Nets does not use activation functions, its performance will degrade in larger networks. We have conducted experiments to compare our method to $\Pi$-Nets on image classification for Cifar 10 and Cifar 100, as shown in the following table. The experimental results show that our method performs better than $\Pi$-Nets in terms of classification accuracy. Second, we developed a novel Taylor initialization to improve the training stability while $\Pi$-Nets may suffer from instability as mentioned in their extension work~[Chrysos 2020]. Lastly, $\Pi$-Nets can be viewed as a special case of TaylorNet at expansion point 0 since the adopted CP-decomposition in $\Pi$-Net is a special case of Tucker decomposition in our work. \
>
> Table: Image classification accuracy for different methods
>  | Model Base 32 |  CIFAR10  |  CIFAR100 |  # of parameters  (M) |
>  |-------------------------------|:-----:|:-----:|:-----:|
> | MLP-Mixer |  94.16$\pm$0.16  |  76.30$\pm$0.25|  59.6   |
> | $\Pi$-Net |  88.12$\pm$0.02  |  67.83$\pm$0.032 |  37  |
> | TaylorNet |  94.97$\pm$0.33  |  79.00$\pm$0.40 |  34  |
>  |         |          |
>
> **References:** \
> [Chrysos 2020] Chrysos, Grigorios G., et al. "P-nets: Deep polynomial neural networks." Proceedings of the IEEE/CVF Conference on Computer Vision and Pattern Recognition. 2020.
>
>
> **Q3.2: More recently, similar ideas are discussed in the tensor community, such as (Qiu et al., 2021 and Rabusseau et al., 2019). In these works, they discussed multi-linear functions as the learning model.**  \
> **A3.2:** Per your suggestion, we have discussed them in the related work of our manuscript.
>
> **Q3.3: The proposed initialization trick is also not novel. A similar idea has been discussed by Pan et al. (2022), which is also not cited in the paper**.  \
> **A3.3:** As suggested, we have added the other initialization methods in the related work in our revised version.
>
> **References:** \
> Pan, Yu, et al. "A Unified Weight Initialization Paradigm for Tensorial Convolutional Neural Networks." International Conference on Machine Learning. PMLR, 2022.
>
> **Q3.4: About the superior performance by the new initialization is ONLY evaluated for the proposed TaylorNet. It is a crucial precondition for the statement, but it is not clearly mentioned when summarising the contributions**. \
> **A3.4:** We have rewritten the contribution of Taylor initialization in our revised manuscript as follows. ``Meanwhile, our Taylor initialization can reach an accuracy that is over 10\% higher than the Xavier and Kaiming initialization methods for the proposed TaylorNet.

---

> > ### Comment · Area_Chair_tBp9 · 2022-11-22
> > **Following up on the rebuttal**
> >
> > Dear reviewer,
> >
> > the authors addressed your points, most notably regarding the novelty of the work and how does it compare with previous polynomial-type neural networks. Are you convinced or otherwise have your opinion influenced?
> >
> > Cheers,
> > Your AC

---

> > > ### Comment · Reviewer_D6Yc · 2022-12-03
> > > **I appreciate the response from the authors, but it does not convince me well.**
> > >
> > > I agreed with the response from the authors regarding the differences from the old polynomial net: the scaling capability of those old polynomial networks is terrible due to the exponential growth of the number of parameters. However, this problem can now be resolved trivially by low-rank tensor representations, as most of the tensor-based learning models did. Based on this point, the novelty of this paper is lower than the average I reviewed at the same level conferences.
> > >
> > > I also appreciate that promised modification of the manuscript by the authors. I will up my recommendation score because of the potential improvement of the manuscript's quality.
> > >
> > > best,

---

### Official Review · Reviewer_LGk5 · 2022-10-20

**Confidence:** 4
**Correctness:** 3
**Technical Novelty And Significance:** 2
**Empirical Novelty And Significance:** 2
**Recommendation:** 3

**Clarity, Quality, Novelty And Reproducibility:**

The Taylor polynomial in eq. (1) is a bit unusual for me. Typically in a Taylor polynomial, there are the higher-order derivatives of the function f at $x_0$, right? Could the authors clarify where this definition without the mentioned derivatives originates from?

There is a significant part that I am not certain after studying the manuscript and I would appreciate the authors’ clarification. The paper is mentioned as “TaylorNet” and introduces “inductive bias to DNNs with Taylor series expansion”. Where are the derivatives computed?

Continuing in the same line as above, why are the $\mathbf{\mathcal{W}}^{[k]}$ mentioned as derivatives? Aren’t they optimized with SGD (or a variant)?


In sec. 2 the paper mentions that $\Pi$-nets are a special form of TaylorNet at $x_0=0$. In sec. 5.2 the paper mentions that the expansion of $x_0=0$ is used for the proposed method. That would mean that the method named as Taylor-Mixer is a $\Pi$-net in reality. Could the authors elaborate on that?

Continuing in the logic mentioned above, the related works of polynomial nets have not been included in the comparisons, which makes it unclear how the proposed method fares with respect to the existing polynomial nets.

This is not the first time an initialization has been proposed for polynomial expansions. The ReLinear initialization has been introduced in “On Expressivity and Training of Quadratic Networks”; could the authors include this in their comparisons in sec. 5.1?

This work is motivated by mentioning that physical priors are important in ML. However, I am not certain what the physical prior is in this case. Could the authors elaborate on this point?

What is the “!!” symbol used in sec. 4.3? It is not defined in the main paper (or the supplementary table).

One suggestion beyond the parameters is to include the number of FLOPs. I expect those might be more than the MLPMixer, or even $\Pi$-nets, however this provides a more complete idea to the reader.

A number of works have also considered polynomial nets for tackling the task, for instance you can find several in the “Augmenting Deep Classifiers with Polynomial Neural Networks” paper. It would be recommended to discuss those related works in more detail. Additionally, this paper identifies that the Taylor polynomials introduce a different inductive bias, however there are already works who have been exploring the inductive bias from a theoretical perspective.

What are the ranks used in each Tucker decomposition from the input to the output of the final network? It would provide further insights to the readers to explain and ablate this point.

The paper requires proofreading, since there are a number of issues currently:
* The \citep{} command needs to be used appropriately. For instance, in sec. 2 “Tensor decomposition Kolda and Bader (2009) aims [...]”; change to citep. In the same paragraph several instances of this exist.
* “and 5) Our approach”: why is the O capital?
* The notation mentions that the matrices are denoted with uppercase boldface letters, however $\mathbf{\mathcal{X}}_{(n)}$ is with calligraphic letter, etc.
* The notation could be simpler. For instance, in sec. 4.1 the $k$ is used as the order of the polynomial expansion, while in sec. 4.2 as an index. Keeping track of all the variables becomes harder, so it would be easier to use different variable names.

I am not certain how eq. (4) and eq. (3) which are purely about tensors dictate how to move from eq. (8) to eq. (9), so some further explanation in the manuscript would make it easier for the reader.

Lastly, I suggest to the authors to include a Lemma of how to go from the two mode-m products of eq. (6) to the product of eq. (7). It is not trivial to realize the first one is not contracting the tensor, since it's a mode-m product with a matrix.


**Strength And Weaknesses:**

 * [+] Improving the generalization in models that non-black box DNNs is important.

 * [+] Using the Tucker decomposition in a smart way in Mixer-like architectures is new to me, even if the Tucker decomposition and polynomial expansions have been used in the past in vision and deep learning approaches.

 * [-] Several technical parts are unclear, while the paper requires proofreading to make it more understandable.

 * [-] The empirical evaluation is weak. The MLP-Mixer considered as a baseline is evaluated both on Imagenet, and VTA-1k which includes 19 datasets. Similarly, $\Pi$-nets that are considered another baseline include more experiments in classification, generation, or mesh representation learning.

 * [-] The literature review on both Tucker decomposition and its use in deep learning and the polynomial nets could be significantly updated. (see below)


**Summary Of The Paper:**

The paper introduces a new architecture that relies on polynomial expansions. The idea is to use polynomial layers, where each layer expresses a polynomial expansion of order $k$ (where $k=2$ in practice). The new architecture uses a Tucker decomposition to reduce the learnable parameters. The authors also introduce a variant called reducible-TaylorNet, where the parameters are further reduced using a re-parametrization. The architecture is validated in small image datasets (CIFAR10/100), two dynamical systems and a small scale NLP task.

**Summary Of The Review:**

The idea of using polynomial expansions is interesting, while the use of the Tucker decomposition to reduce the parameters can be a useful tool. However, the paper currently has several flaws as identified above, making it hard for the reader to appreciate the contribution. At the same time, the experiments are weak. Nevertheless, if the revised version addresses all of the concerns, I am willing to reconsider my rating.

---

> ### Author Response · Authors · 2022-11-19
> **Response to Reviewer LGk5 (1/2)**
>
> **Q2.1: The Taylor polynomial in eq. (1) is a bit unusual for me. Typically in a Taylor polynomial, there are the higher-order derivatives of the function f at $x_0$, right? Could the authors clarify where this definition without the mentioned derivatives originates from?** \
> **A2.1:** Thanks for the comments! According to the textbook [Thomas 2005], we have rewritten the Taylor Polynomials for a vector-valued multivariate function in Eq.1.
>
> **References:** \
> [Thomas 2005] George Brinton Thomas, Maurice D Weir, Joel Hass, and Frank R Giordano. Thomas’ calculus.
> Addison-Wesley, 2005
>
> **Q2.2: There is a significant part that I am not certain after studying the manuscript and I would appreciate the authors’ clarification. The paper is mentioned as “TaylorNet” and introduces “inductive bias to DNNs with Taylor series expansion”. Where are the derivatives computed? $\boldsymbol{W}$ are derivatives? Aren’t they optimized with SGD (or a variant)?** \
> **A2.2:** Since Taylor series expansion removes some redundant links between different variables in deep neural networks, it may introduce the inductive bias from the math perspective. In addition, prior work~[Grigorios 2022], recommended by the Reviewer, has pointed out that polynomial neural networks can introduce inductive bias. In addition, $\mathcal{W}$ denotes partial derivatives in the original Taylor polynomial in math. Since the mapping function is unknown and needs to be learned in our work, the derivatives, $\mathcal{W}$, are learnable parameters by DNNs and will be optimized during back propagation.
>
> **References:**  \
> [Grigorios 2022] Augmenting Deep Classifiers with Polynomial Neural Networks, 2022.
>
>
> **Q2.3: In sec. 2 the paper mentions that $\Pi$-nets are a special form of TaylorNet at $x_0=0$. In sec. 5.2 the paper mentions that the expansion of 0 is used for the proposed method. That would mean that the method named as Taylor-Mixer is a Pi-net in reality. Could the authors elaborate on that?** \
> **A2.3:** In reality, we would like to clarify that the Taylor expansion point is the mean value of the input data. Since data normalization technique is used in data pre-processing, we claim that the expansion is 0 in the experiments. However, we have verified that our TaylorNet still works at the other expansion points.
>
> **Q2.4: On comparing the proposed method with the existing polynomial nets.** \
> **A2.4**: We have conducted experiments to compare our method with the baseline $\Pi$-net. The results in the following table show that our method can achieve a higher accuracy than $\Pi$-net for CIFAR10 and CIFAR100
>
>  | Model Base 32 |  CIFAR10  |  CIFAR100 |  # of parameters  (M) |
> |-------------------------------|:-----:|:-----:|:-----:|
>  | MLP-Mixer |  94.16$\pm$0.16  |  76.30$\pm$0.25|  59.6   |
>  | $\Pi$-Net |  88.12$\pm$0.02  |  67.83$\pm$0.032 |  37  |
>  | TaylorNet |  94.97$\pm$0.33  |  79.00$\pm$0.40 |  34  |
> |         |          |
>
>
> **Q2.5: This is not the first time an initialization has been proposed for polynomial expansions. The ReLinear initialization has been introduced in “On Expressivity and Training of Quadratic Networks”; could the authors include this in their comparisons in sec. 5.1?** \
> **A2.5:** Due to limited rebuttal time, we will leave the comparison of ReLinear initialization to future work.

---

> > ### Author Response · Authors · 2022-11-19
> > **Response to Reviewer LGk5 (2/2)**
> >
> > **Q2.6: This work is motivated by mentioning that physical priors are important in ML. However, I am not certain what the physical prior is in this case. Could the authors elaborate on this point?**  \
> > **A2.6:** In this work, we think Taylor series expansion may introduce physical prior to the TaylorNet since it can remove some redundant links among different variables (neurons) in the conventional MLP. This has been studied by the prior work in Phy-Taylor[Mao et al 2022]. In addition, our TaylorNet does not use any activation functions, it allows us to incorporate physics knowledge into the model to introduce physical prior.
> >
> > **References:** \
> > [Mao et al 2022] Mao, Yanbing, et al. "Phy-Taylor: Physics-Model-Based Deep Neural Networks." arXiv preprint arXiv:2209.13511 (2022).
> >
> > **Q2.7: What is the “!!” symbol used in sec. 4.3? It is not defined in the main paper (or the supplementary table).** \
> > **A2.7**: “!!” symbol denotes the double factorial in our work. We have added its definition to our revised manuscript.
> >
> > **Q2.8: One suggestion beyond the parameters is to include the number of FLOPs. I expect those might be more than the MLPMixer, or even $\Pi$-nets, however this provides a more complete idea to the reader.** \
> > **A2.8:** Thanks for the suggestion! We will add the FLOPs in our final version.
> >
> > **Q2.9: A number of works have also considered polynomial nets for tackling the task, for instance you can find several in the “Augmenting Deep Classifiers with Polynomial Neural Networks” paper. It would be recommended to discuss those related works in more detail. Additionally, this paper identifies that the Taylor polynomials introduce a different inductive bias, however there are already works who have been exploring the inductive bias from a theoretical perspective.** \
> > **A2.9:** Per your suggestion, we have discussed this paper in the related work.
> >
> >
> > **Q2.10: What are the ranks used in each Tucker decomposition from the input to the output of the final network? It would provide further insights to the readers to explain and ablate this point.** \
> > **A2.10:** We have introduced the rank used in our experiments in Appendix C. We describe the rule of thumb for choosing the ranks as follows. For a $N$-th order Taylor layer with input and output rank $r_{\text{in},k,j}$ and $r_{\text{out},k}$, the effective width of this layer is approximately $\min(\sum_{k=1}^N \sum_{j=1}^k  j r_{\text{in},k,j}, \sum_{k=1}^N  r_{\text{out},k})$. Therefore, in order to achieve larger width with fixed number of parameters, we should set $\sum_{k=1}^N \sum_{j=1}^k  j r_{\text{in},k,j} \approx \sum_{k=1}^N  r_{\text{out},k}$. Then we can scale $r_{\text{in},k,j}$ and $r_{\text{out},k}$ together to adjust the number of parameters of the model.
> >
> > **Q2.11: I am not certain how eq. (4) and eq. (3) which are purely about tensors dictate how to move from eq. (8) to eq. (9), so some further explanation in the manuscript would make it easier for the reader.** \
> > **A2.11**: According to the highly-cited work~[Kolda 2009], we can transform the tensor in Eq. 3 into factorized matrices in Eq. 4 based on
> > mode-$n$ product. Using the same methodology, we can convert from Eq. 8 to Eq.9. We have provided detailed explanation in our revised version.
> >
> > **References:** \
> > [Kolda 2009] Kolda, T. G., \& Bader, B. W. (2009). Tensor decompositions and applications. SIAM review, 51(3), 455-500.
> >
> > **Q2.12: Lastly, I suggest to the authors to include a Lemma of how to go from the two mode-m products of eq. (6) to the product of eq. (7). It is not trivial to realize the first one is not contracting the tensor, since it's a mode-m product with a matrix.** \
> > **A2.12:** As suggested, we have included two Lemmas about converting from Eq. 6 to Eq. 8 in the appendix and further present the detailed transformation based on the two Lemmas.

---

> > > ### Comment · Reviewer_LGk5 · 2022-11-19
> > > **Further remarks, e.g. with respect to the name**
> > >
> > > I am thankful to the authors for the detailed responses.
> > >
> > > The responses state that the "function is unknown and needs to be learned in our work, the derivatives, W, are learnable parameters by DNNs and will be optimized during back propagation.".
> > > If the derivatives $W$ are learned, then this is not the Taylor expansion. As such, the naming of the proposed method is unclear. Similarly, the relationship with $\Pi$-net is unclear, since it was previously mentioned as the Taylor expansion around the point $0$.
> > >
> > > Unless there is proof that the $W$ are indeed derivatives, I would not recommend claiming they are derivatives in the manuscript, as this is confusing to the reader.
> > >
> > > In addition, the responses (to other reviewers) mention that the initialization scheme is one key contribution in this work. However, the differences from previously proposed initializations (e.g. ReLinear) for polynomial nets is unclear. Also, there is no such comparison to date between the initializations.
> > >
> > > What is more, the experimental evaluation is still weak, for instance there is no experiment on ImageNet to show how the proposed methods fare with respect to MLPMixer and $\Pi$-nets.

---

### Official Review · Reviewer_9i73 · 2022-10-25

**Confidence:** 4
**Correctness:** 3
**Technical Novelty And Significance:** 2
**Empirical Novelty And Significance:** 1
**Recommendation:** 5

**Clarity, Quality, Novelty And Reproducibility:**

- The paper is clear for the most part. The quality of the writing is adequate.

- While novelty is a subjective concept, I believe that apart from the TaylorNet specific weight initialization, the rest of the concepts are already prevalent in the literature in one form or the other. From that perspective, in my opinion, the contributions of the paper are limited.

- Unless I missed it, the paper does not propose to release code publicly. While the image classification and sentiment analysis experiments might be reproducible from the hyper-parameter details in the supplementary, I think the system identification experiments cannot be reproduced.

**Strength And Weaknesses:**

Strengths:
1. The custom Taylor initialization is interesting, especially in the context of activation free networks.
2. The concept of activation free networks is interesting, although its practical utility is unclear.
3. Utility of TaylorNet is evaluated across a diverse set of tasks, image classification, sentiment analysis and dynamical systems.

Weaknesses:
1. The motivation for the work is not clear, especially given the existence of $\Pi$-Nets. As far as I know, $\Pi$-Nets are very similar, one can use 2nd order approximation in $\Pi$-Nets and there is a version of it without activation functions as well.
2. The motivation for networks without activation functions is also not very clear. TaylorNet is a composite function of TaylorBlocks. While TaylorBlocks themselves are interpretable, TaylorNet itself is too complex to interpret. The motivation is even less clear in tasks like image classification.
3. Experimental comparisons to important baselines are missing. For example, $\Pi$-Nets [1] in the case of image classification, SINDy [2] and its variants in the case of system identification for dynamical systems.
4. Objectively the results are middling at best. There are a plethora of alternative CNN based architectures that achieve similar or better performance on CIFAR datasets compared to the results presented here. The advantage presented by TaylorNet over such models is unclear.

Other Comments:
- The claim about interpretability is too strong. The dynamical systems example is too simplistic.
- In the experiment in 5.1, the conclusion that Taylor initialization is better than Xavier or Kaiming initialization is drawn from a simple two-layer network. Can you comment on whether this is expected to hold over larger/deeper networks as well?

[1] $\Pi$-nets: Deep polynomial neural networks, CVPR 2020
[2] Discovering governing equations from data by sparse identification of nonlinear dynamical systems, PNAS 2016

**Summary Of The Paper:**

The paper proposes a neural architecture called TaylorNet. It is designed to mimic a taylor-series approximation of a multivariate function. The network itself is expressed as a composite function of Taylor layers, where each layer is a 2nd order taylor expansion. A key characteristic of the model is that it is free of non-linear activation functions, and the paper also proposes a customized weight initialization method. Experiments are performed on three tasks: image classification (CIFAR-10 and CIFAR-100), dynamical systems (Duffing Equation, Non-Linear Fluid Flow), and sentiment analysis (IMDB).

**Summary Of The Review:**


Overall, the paper presents an interesting architecture based on the idea of Taylor approximation of multivariate functions. However, a majority of the conceptual contributions in this paper already exist in the literature. From a practical perspective, the contribution of TaylorNet over existing architectures is unclear.

Post Rebuttal Update:

I read the revised paper, comments from other reviewers, and the author's responses. The rebuttal addressed some concerns, especially related to experimental comparisons to $\Pi$-Nets and SINDy. I increased the rating for the paper, but there are still some outstanding concerns.

---

> ### Author Response · Authors · 2022-11-19
> **Response to Reviewer 9i73 (1/2)**
>
> **Q1.1: The motivation for the work is not clear, especially given the existence of Pi-Nets. As far as I know, Pi-Nets are very similar, one can use 2nd order approximation in Pi-Nets and there is a version of it without activation functions as well.**
>
> **A1.1**: The motivation of this work is to develop a genetic neural architecture that can be widely used in various applications, such as scientific discovery, computer vision, and natural language processing.
> We would like to point out the significant difference between our TaylorNet and $\Pi$-Net in three-fold. First of all, while $\Pi$-Nets does not use activation functions, its performance will degrade in larger networks. We have conducted experiments to compare our method to $\Pi$-Nets on image classification for Cifar 10 and Cifar 100. The experimental results in the following Table 1 show that our method performs better than $\Pi$-Nets in terms of classification accuracy. Second, we developed a novel Taylor initialization to improve the training stability while $\Pi$-Nets may suffer from instability as mentioned in their extension work~[Chrysos 2020]. Lastly, $\Pi$-Nets can be viewed as a special case of TaylorNet at expansion point 0 since the adopted CP-decomposition in $\Pi$-Net is a special case of Tucker decomposition in our work.
>
> Table 1: Accuracy comparison for different methods
>
>  | Model Base 32 |  CIFAR10  |  CIFAR100 |  # of parameters  (M) |
> |-------------------------------|:-----:|:-----:|:-----:|
>  | MLP-Mixer |  94.16$\pm$0.16  |  76.30$\pm$0.25|  59.6   |
>  | $\Pi$-Net |  88.12$\pm$0.02  |  67.83$\pm$0.032 |  37  |
>  | TaylorNet |  94.97$\pm$0.33  |  79.00$\pm$0.40 |  34  |
> |         |          |
>
> **References**: \
> [Chrysos 2020]  Chrysos, Grigorios G., et al. "P-nets: Deep polynomial neural networks." Proceedings of the IEEE/CVF Conference on Computer Vision and Pattern Recognition. 2020.
>
>
> **Q1.2: The motivation for networks without activation functions is also not very clear. TaylorNet is a composite function of TaylorBlocks. While TaylorBlocks themselves are interpretable, TaylorNet itself is too complex to interpret. The motivation is even less clear in tasks like image classification**\
> **A1.2:** The key motivation for TaylorNet without activation functions is that it can explicitly learn the mathematical relation between the input $\boldsymbol{x}$ and output $\boldsymbol{y}$ with a smooth polynomial function. It is thus be applied to data-driven scientific discovery that requires to uncover a polynomial relation between the input and output. In Section , we illustrated that TaylorNet can explicitly learn the dynamical systems with Taylor polynomials based on observational data. For instance, our method can successfully discover the ODEs for fluid flow system only based on observed trajectories, as shown in Eq. 21. Furthermore, we can extend our TaylorNet to physics-informed TaylorNet, which aims to incorporate physics knowledge into neural networks to improve model generalization and interpretability. The applications of computer vision include tumor segmentation [Kevin2020] and real-time Human motion tracking [Xinyu 2022]
>
> **References**:\
> [Kevin 2020] "A physics-guided modular deep-learning based automated framework for tumor segmentation in PET." Physics in Medicine \& Biology 65.24 (2020): 245032.  \
> [Xinyu 2022] "Physical Inertial Poser (PIP): Physics-aware Real-time Human Motion Tracking from Sparse Inertial Sensors." Proceedings of the IEEE/CVF Conference on Computer Vision and Pattern Recognition. 2022.

---

> > ### Author Response · Authors · 2022-11-19
> > **Response to Reviewer 9i73 (2/2)**
> >
> > **Q1.3: Experimental comparisons to important baselines are missing. For example, $\Pi$-Nets in the case of image classification, SINDy [2] and its variants in the case of system identification for dynamical systems.** \
> > **A1.3:** We have compared the proposed TaylorNet with the baselines suggested by the reviewer. First, we compare the proposed method with $\Pi$-Nets on image classification using CIFAR10 and CIFAR100, as shown in the above Table 1. We can see that our method outperforms $\Pi$-Nets on image classification.
> >
> > Second, we compare the performance of our method and SINDy on dynamical systems, including Duffing and fluid flow dynamics. Below, we illustrate the discovered ODEs by SINDy using PySINDy [Brian 2020, Kaptanoglu 2022]on GitHub (https://github.com/dynamicslab/pysindy).
> > The learned Duffing equation by SINDy is \
> > $\dot{x}_1 =  x_1$ \
> > $\dot{x}_2 = 1.319 \sin x_1 -0.311 x_1 + 0.035 x_1^2 -0.027 x_2^2 -0.777 x_1^3 + 0.013 x_1 x_2^2$ \
> > We can see that SINDy performs worse than our TaylorNet in predicting Duffing equation. The main reason is that SINDy is based on a lot of candidate functions during learning, which may interfere with each other. In addition, another limitation of SINDy is that it only works for a single trajectory rather than multiple trajectories. In contrast, the TaylorNet can approximate any smooth non-linear functions under real interval.
> >
> > **References**: \
> > [Brian 2020] Brian M. de Silva, Kathleen Champion, Markus Quade, Jean-Christophe Loiseau, J. Nathan Kutz, and Steven L. Brunton., (2020). PySINDy: A Python package for the sparse identification of nonlinear dynamical systems from data. Journal of Open Source Software, 5(49), 2104. \
> > [Kaptanoglu 2022] Kaptanoglu et al., (2022). PySINDy: A comprehensive Python package for robust sparse system identification. Journal of Open Source Software, 7(69), 3994.\
> >
> > **Q1.4: There are a plethora of alternative CNN based architectures that achieve similar or better performance on CIFAR datasets compared to the results presented here. The advantage presented by TaylorNet over such models is unclear.** \
> > **A1.4**: We compared our Taylor-Mixer with MLP-Mixer since the latter can achieve comparable accuracy to ViT and better than ResNet and HaloNet, as mentioned in the original work~[Tolstikhin 2021]. In this work, we illustrated our method performs better than MLP-Mixer with less parameters, which means that our method can outperform some CNN based architectures, such as ResNet and HaloNet. Nevertheless, we will add more experiments to compare some state-of-the-art CNNs in our final version.
> >
> > **References:** \
> > [Tolstikhin 2021] Tolstikhin, Ilya O., et al. "Mlp-mixer: An all-mlp architecture for vision." Advances in Neural Information Processing Systems 34 (2021): 24261-24272.
> >
> > **Q1.5: The claim about interpretability is too strong. The dynamical systems example is too simplistic.** \
> > **A1.5**: We changed the claim that our method can enable interpretability in dynamical systems since it can explicitly discover the ODEs based on observational data. To further demonstrate the power of the proposed TaylorNet, we applied it to identify Lorenz model, which is an unstable oscillatory system. The learned Lorenz equations are given by \
> > $\dot{x}_1 = 9.9376x_2 - 9.8122x_1$ \
> > $\dot{x}_2 = 27.9305x_1 - 0.8752 x_2 - 0.9948 x_1 x_3 $ \
> > $\dot{x}_3 = 0.9908x_1 x_2 - 2.9345x_3.$ \
> > which is very close to the ground-truth equations \
> > $\dot{x}_1 = 10(x_2 - x_1),$ \
> > $ \dot{x}_2 = 30x_1 - x_2 - x_1 x_3, $ \
> > $\dot{x}_3 = x_1 x_2 - 3x_3.$
> >
> > **Q1.6: In the experiment in 5.1, the conclusion that Taylor initialization is better than Xavier or Kaiming initialization is drawn from a simple two-layer network. Can you comment on whether this is expected to hold over larger/deeper networks as well?** \
> > **A1.6:** Yes. We have conducted experiments to validate that it still works for larger networks. Specifically, we use 4-layer Taylor-Mixer (34.4M parameters) to compare the performance of our method with Xavier or Kaiming initialization. The results, as shown in Fig. 2 in our revised version, demonstrate that our method can achieve over 10\% higher accuracy than Xavier or Kaiming initialization.

---

> > > ### Comment · Area_Chair_tBp9 · 2022-11-22
> > > **Following up the rebuttal**
> > >
> > > Dear reviewer,
> > >
> > > the authors tried to address your points, especially commenting on the similarity with \Pi-nets, saying that they do not generalize well with deeper networks. Overall, how has the response influenced your opinion?
> > >
> > > Cheers,
> > > Your AC

---

### Author Response · Authors · 2022-11-19
**General Response to Reviewers**

We thank the reviewers for their constructive comments and thorough reviews. We have responded to each reviewer's comments below and have uploaded a revised version of the manuscript. In particular, we have conducted experiments to compare our method to $\Pi$-Nets on image classification for CIFAR10 and CIFAR100. The results show that our method can achieve a higher accuracy than $\Pi$-nets .Please let us know if you have any additional concerns. I am looking forward to your reply!

---

### Decision · Program_Chairs · 2023-01-20

**Decision:**

Reject

**Justification For Why Not Higher Score:**

Overlclaiming what the method is supposed to be doing is an obvious red flag, see summary for details.

**Justification For Why Not Lower Score:**

The problem definition is interesting.

**Metareview: Summary, Strengths And Weaknesses:**

The submission proposes TaylorNets, that is neural networks that learn higher-order Taylor polynomial expansions to approximate the functions to be learned from data, $f: X \rightarrow Y$. The proposed algorithm has some interesting contributions acknowledged by reviewers, importantly the Tucker decomposition, to reduce learnable parameters. However, as the authors also acknowledge, the variables that are supposed to be the derivatives of the Taylor expansion are in fact learnable parameters. Thus, there is no evidence that these correspond to derivatives, other than it 'is supposed to be like that because of the polynomial form'. This is simply not good enough and certainly overclaiming. On that ground alone, the paper cannot be accepted. Besides that, it is unclear to what extent the algorithm leads to interpretable results, as well as what is the precise difference with $Pi$-Nets, if in the end the the decomposition is centered around $x_0$.